# Zero-One Laws of Graph Neural Networks

**Sam Adam-Day**[*]
Department of Mathematics
University of Oxford
Oxford, UK
`sam.adam-day@cs.ox.ac.uk`

**Theodor-Mihai Iliant**
Department of Computer Science
University of Oxford
Oxford, UK
`theodor-mihai.iliant@lmh.ox.ac.uk`

**İsmail İlkan Ceylan**
Department of Computer Science
University of Oxford
Oxford, UK
`ismail.ceylan@cs.ox.ac.uk`

## Abstract

Graph neural networks (GNNs) are the de facto standard deep learning architectures for machine learning on graphs. This has led to a large body of work analyzing the capabilities and limitations of these models, particularly pertaining to their representation and extrapolation capacity. We offer a novel theoretical perspective on the representation and extrapolation capacity of GNNs, by answering the question: how do GNNs behave as the number of graph nodes become very large? Under mild assumptions, we show that when we draw graphs of increasing size from the Erdős-Rényi model, the probability that such graphs are mapped to a particular output by a class of GNN classifiers tends to either *zero* or to *one*. This class includes the popular graph convolutional network architecture. The result establishes 'zero-one laws' for these GNNs, and analogously to other convergence laws, entails theoretical limitations on their capacity. We empirically verify our results, observing that the theoretical asymptotic limits are evident already on relatively small graphs.

## 1 Introduction

Graphs are common structures for representing relational data in a wide range of domains, including physical [35], chemical [7, 18], and biological [42, 10] systems, which sparked interest in machine learning over graphs. Graph neural networks (GNNs) [33, 14] have become prominent models for graph machine learning for a wide range of tasks, owing to their capacity to explicitly encode desirable relational inductive biases [5]. One important virtue of these architectures is that every GNN model can be applied to *arbitrarily large* graphs, since in principle the model parameters are independent of the graph size. This raises the question: how do GNNs behave as the number of nodes becomes very large? When acting as binary classifiers, GNNs can be thought of as parametrizing Boolean properties of (labelled) graphs. A classical method of specifying such properties is through first-order formulas, which allow for precise definitions using a formal language [9]. The celebrated 'zero-one law' for first-order logic [13, 8] provides a crisp answer to the question of the asymptotic behaviour of such properties: as graphs of increasing size are drawn from the Erdős-Rényi distribution, the probability that a *first-order* property holds either tends to *zero* or to *one*.

---

[*]Alternative email address: `me@samadamday.com`

37th Conference on Neural Information Processing Systems (NeurIPS 2023).

In this paper, we show an analogous result for binary classification GNNs: under mild assumptions on the model architecture, several GNN architectures including graph convolutional networks [21] satisfy a zero-one law over Erdős-Rényi graphs with random node features. The principal import of this result is that it establishes a novel upper-bound on the expressive power of GNNs: any property of graphs which can be *uniformly* expressed by a GNN must obey a zero-one law. An example of a simple property which does *not* asymptotically tend to zero or one is that of having an even number of nodes. Note however that our result, combined with the manifest success of GNNs in practice, suggests that zero-one laws must be abundant in nature: if a property we cared about did not satisfy a zero-one law, none of the GNN architectures we consider would be able to express it.

Our main results flexibly apply both to the case where we consider the GNN as a classifier applied to randomly sampled graphs and node features, and where we consider the random node features as part of the model. For the latter, it is known that incorporating randomized features into the model significantly increases its expressive power on graphs with *bounded number of nodes* [32, 1]. Our results yield the first upper bound on the expressive power of these architectures which is *uniform* in the number of nodes. We complement this with a corresponding lower bound, showing that these architectures can universally approximate any property which satisfies a certain zero-one law.

A key strength of the results is that they apply equally well to randomly initialized networks, trained networks, and anything in between. In this sense, our asymptotic analysis is orthogonal to the question of optimization, and holds regardless of the choice of training method. Another interesting aspect of these results is that they unite analysis of expressive power with extrapolation capacity. Our zero-one laws simultaneously provide limits on the ability of GNNs to extrapolate from smaller Erdős-Rényi graphs to larger ones: eventually any GNN must classify all large graphs the same way.

To validate our theoretical findings, we conduct a series of experiments: since zero-one laws are of asymptotic nature, we may need to consider very large graphs to observe clear empirical evidence for the phenomenon. Surprisingly however, GNNs already exhibit clear evidence of a zero-one law even on small graphs. Importantly, this is true for networks with very few layers (even a single-layer), which is reassuring, as it precludes confounding factors, such as the effect of over-smoothing due to increased number of layers [23]. We provide further experimental results in the appendix of this paper, where all proofs of technical statements can also be found. We make the code for our experiments available online at `https://github.com/SamAdamDay/Zero-One-Laws-of-Graph-Neural-Networks`.

## 2  Preliminaries

**Random graphs and matrices.** The focus of our study is on classes of random graphs with random features, for which we introduce some notation. We write $\boldsymbol{x} \in \mathbb{R}^d$ to represent a vector, and $\boldsymbol{X} \in \mathbb{R}^{d \times n}$ to represent a matrix. Analogously, we write $\mathbf{x}$ to denote a *random* vector, and $\mathbf{X}$ to denote a *random* matrix, whose entries are (real) random variables. We write $\mathbb{G}(n, r)$ to denote the class of simple, undirected Erdős-Rényi (ER) graphs with $n$ nodes and edge probability $r$ and let $\mathbb{D}(d)$ denote some distribution of feature vectors over $\mathbb{R}^d$. We define an Erdős-Rényi graph equipped with random node features as a pair $\mathcal{G} = (\mathbf{A}, \mathbf{X})$, where $\mathbf{A} \sim \mathbb{G}(n, r)$ is the random graph adjacency matrix of the graph $G = (V, E)$ and $\mathbf{X} \in \mathbb{R}^{d \times n}$ is a corresponding random feature matrix, independent of $G$, which contains, for each node $v \in V$, an initial random node feature $\mathbf{x}_v \sim \mathbb{D}(d)$ as the corresponding columns of $\mathbf{X}$.[2]

**Message passing neural networks.** The focus of this work is on *message-passing neural networks (MPNNs)* [12, 17] which encapsulate the vast majority of GNNs. The fundamental idea in MPNNs is to update the initial (random) state vector $\mathbf{x}_v^{(0)} = \mathbf{x}_v$ of each node $v$ for $T \in \mathbb{N}$ iterations, based on its own state and the state of its neighbors $\mathcal{N}(v)$ as:

$$\mathbf{x}_v^{(t+1)} = \phi^{(t)}\Big(\mathbf{x}_v^{(t)}, \psi^{(t)}\big(\mathbf{x}_v^{(t)}, \{\!\!\{\mathbf{x}_u^{(t)} \,|\, u \in \mathcal{N}(v)\}\!\!\}\big)\Big),$$

where $\{\!\!\{\cdot\}\!\!\}$ denotes a multiset, and $\phi^{(t)}$ and $\psi^{(t)}$ are differentiable *combination*, and *aggregation* functions, respectively. Each layer's node representations can have different dimensions: we denote by $d(t)$ the dimension of the node embeddings at iteration $t$ and typically write $d$ in place of $d(0)$.

---

[2]We define a $d \times |V|$ dimensional (random) feature matrix as opposed to the more common $|V| \times d$. This is for ease of presentation, since we aim to work on the (random) column vectors of such matrices.

The final node representations can then be used for node-level predictions. For graph-level predictions, the final node embeddings are *pooled* to form a graph embedding vector to predict properties of entire graphs. The pooling often takes the form of simple averaging, summing or component-wise maximum. For Boolean node (resp., graph) classification, we further assume a classifier $\mathfrak{C} : \mathbb{R}^{d(T)} \to \mathbb{B}$ which acts on the final node representations (resp., on the final graph representation).

There exist more general message passing paradigms [5] such as *MPNNs with global readout* which additionally aggregate over all node features at every layer, and are known to be more expressive [4]. Some model architectures considered in this paper include a global readout component and we consider different choices for the combine ($\phi^{(t)}$) and aggregate ($\psi^{(t)}$) functions, as we introduce next.

**GCN.** The primary GNN architecture we consider is *graph convolutional networks* (GCN) [21]. These are instances of MPNNs with self-loops, which aggregate over the extended neighborhood of a node $\mathcal{N}^+(v) := \mathcal{N}(v) \cup \{v\}$. GCNs iteratively update the node representations as $\mathbf{x}_v^{(t)} = \sigma\left(\mathbf{y}_v^{(t)}\right)$, where the preactivations are given by:

$$\mathbf{y}_v^{(t)} = \boldsymbol{W}_n^{(t)} \sum_{u \in \mathcal{N}^+(v)} \frac{1}{\sqrt{|\mathcal{N}(u)||\mathcal{N}(v)|}} \mathbf{x}_u^{(t-1)} + \boldsymbol{b}^{(t)}$$

We apply the linear transformation $\boldsymbol{W}_n^{(t)} \in \mathbb{R}^{d(t) \times d(t-1)}$ to a normalized sum of the activations for the previous layers of the neighbors of the node under consideration, together with its own activation. Adding a bias term $\boldsymbol{b}^{(t)}$ yields the preactivation $\mathbf{y}_v^{(t)}$, to which we apply the non-linearity $\sigma$.

**MEANGNN.** We also consider the MEANGNN$^+$ architecture which is a *self-loop GNN with mean aggregation and global readout* [17], and updates the node representation as $\mathbf{x}_v^{(t)} = \sigma\left(\mathbf{y}_v^{(t)}\right)$, where:

$$\mathbf{y}_v^{(t)} = \frac{1}{|\mathcal{N}^+(v)|} \boldsymbol{W}_n^{(t)} \sum_{u \in \mathcal{N}^+(v)} \mathbf{x}_u^{(t-1)} + \frac{1}{n} \boldsymbol{W}_r^{(t)} \sum_{u \in V} \mathbf{x}_u^{(t-1)} + \boldsymbol{b}^{(t)}$$

MEANGNN$^+$ models additionally apply a linear transformation $\boldsymbol{W}_r^{(t)} \in \mathbb{R}^{d(t) \times d(t-1)}$ to the mean of all previous node representations. We refer to MEANGNN as the special case of this architecture which does *not* include a global readout term (obtained by dropping the second term in the equation).

**SUMGNN.** Finally, we consider the SUMGNN$^+$ architecture which is a *GNN with sum aggregation and global readout* [12], and updates the node representations as $\mathbf{x}_u^{(t)} = \sigma\left(\mathbf{y}_u^{(t)}\right)$, where:

$$\mathbf{y}_v^{(t)} = \boldsymbol{W}_s^{(t)} \mathbf{x}_v^{(t-1)} + \boldsymbol{W}_n^{(t)} \sum_{u \in \mathcal{N}(v)} \mathbf{x}_u^{(t-1)} + \boldsymbol{W}_r^{(t)} \sum_{u \in V} \mathbf{x}_u^{(t-1)} + \boldsymbol{b}^{(t)}$$

This time, we separate out the contribution from the preactivation of the previous activation for the node itself. This yields three linear transformations $\boldsymbol{W}_s^{(t)}, \boldsymbol{W}_n^{(t)}, \boldsymbol{W}_r^{(t)} \in \mathbb{R}^{d(t) \times d(t-1)}$. The corresponding architecture without the global readout term is called SUMGNN.

## 3 Related work

Graph neural networks are flexible models which can be applied to graphs of any size following training. This makes an asymptotic analysis in the size of the input graphs very appealing, since such a study could lead to a better understanding of the extrapolation capabilities of GNNs, which is widely studied in the literature [41, 40]. Previous studies of the asymptotic behaviour of GNNs have focused on convergence to theoretical limit networks [20, 31] and their stability under the perturbation of large graphs [11, 22].

Zero-one laws have a rich history in first-order logic and random graph theory [13, 8, 25, 34, 6]. Being the first of its kind in the graph machine learning literature, our study establishes new links between graph representation learning, probability theory, and logic, while also presenting a new and interesting way to analyze the expressive power of GNNs. It is well-known that the expressive power of MPNNs is upper bounded by the *1-dimensional Weisfeiler Leman graph isomorphism test (1-WL)* [39, 29] and architectures such as SUMGNN$^+$ [29] can match this. Barceló et al. [4] further gives a logical characterization for a class of MPNNs showing SUMGNN$^+$ can capture any

function which can be expressed in the logic $\mathsf{C}^2$, which is an extension of the two-variable fragment of first-order logic with counting quantifiers. Several works study the expressive power of these models under the assummption that there are *unique node identifiers* [26], or define *higher-order* GNN models [29, 27, 28, 19] to obtain more expressive architectures.

Our work has direct implications on GNNs using random node features [32, 1], which are universal in the bounded graph domain. Specifically, we derive a zero-one law for GNNs using random node features which puts an upper bound on the expressive power of such models in a uniform sense: what class of functions on graphs can be captured by a *single* GNN with random node features? Abboud et al. [1] prove a universality result for these models, but it is not uniform, since the construction depends on the graph sizes, and yields a different model parametrization depending on the choice of the graph sizes. Moreover, the construction of Abboud et al. [1] is of size exponential in the worst case. Grohe [15] recently improved upon this result, by proving that the functions that can be computed by a *polynomial-size bounded-depth* family of GNNs using random node features are exactly the functions computed by bounded depth Boolean circuits with threshold gates. This establishes an upper bound on the power of GNNs with random node features, by requiring the class of models to be of bounded depth (fixed layers) and of size polynomial. However, this result is still *not* uniform, since it allows the target function to be captured by different model parametrizations. There is no known upper bound for the expressive power of GNNs with random node features in the uniform setting, and our result establishes this.

Other limitations of MPNNs include *over-smoothing* [23, 30] and *over-squashing* [2] which are related to information propagation, and are linked to using more message passing layers. The problem of over-smoothing has also been studied from an asymptotic perspective [23, 30], where the idea is to see how the node features evolve as we increase the number of layers in the network. Our study can be seen as orthogonal to this work: we conduct an asymptotic analysis in the size of the graphs rather than in the number of layers.

## 4  Zero-one laws of graph neural networks

### 4.1  Problem statement

We first define graph invariants following Grohe [16].

**Definition 4.1.** A *graph invariant* $\xi$ is a function over graphs, such that for any pair of graphs $G_1$, $G_2$, and, for any isomorphism $f$ from $G_1$ to $G_2$ it holds that $\xi(G_1) = \xi(f(G_2))$. Graph invariants for graphs with node features are defined analogously.

Consider any GNN model $\mathcal{M}$ used for binary graph classification. It is immediate from the definition that $\mathcal{M}$ is invariant under isomorphisms of the graphs on which it acts. Hence $\mathcal{M}$, considered as function from graphs to $\mathbb{B} = \{0, 1\}$, is a graph invariant. In this paper, we study the asymptotic behavior of $\mathcal{M}$ as the number of nodes increases.

One remarkable and influential result from finite model theory is the 'zero-one law' for first-order logic. A (Boolean) graph invariant $\xi$ satisfies a *zero-one law* if when we draw graphs $G$ from the ER distribution $\mathbb{G}(n, r)$, as $n$ tends to infinity the probability that $\xi(G) = 1$ either tends to 0 or tends to 1. The result, due to Glebskii et al. [13] and Fagin [8], states that any graph invariant which can be expressed by a first-order formula satisfies a zero-one law. Inspired by this asymptotic analysis of first-order properties, we ask whether GNNs satisfy a zero-one law. As the input of a GNN is a graph with node features, we need to reformulate the statement of the law to incorporate these features.

**Definition 4.2.** Let $\mathcal{G} = (\mathbf{A}, \mathbf{X})$ be a graph with node features, where $\mathbf{A} \sim \mathbb{G}(n, r)$ is a graph adjacency matrix and, independently, $\mathbf{X}$ is a matrix of node embeddings, where $\mathbf{x}_v \sim \mathbb{D}(d)$ for every node $v$. A graph invariant $\xi$ for graphs with node features satisfies a *zero-one law* with respect to $\mathbb{G}(n, r)$ and $\mathbb{D}(d)$ if, as $n$ tends to infinity, the probability that $\xi(\mathcal{G}) = 1$ tends to either 0 or 1.

Studying the asymptotic behavior of GNNs helps to shed light on their capabilities and limitations. A zero-one law establishes a limit on the ability of such models to extrapolate to larger graphs: any GNN fitted to a finite set of datapoints will tend towards outputting a constant value on larger and larger graphs drawn from the distribution described above. A zero-one law in this setting also transfers to a corresponding zero-one law for GNNs with random features. This establishes an upper-bound on the uniform expressive power of such models.

## 4.2 Graph convolutional networks obey a zero-one law

Our main result in this subsection is that (Boolean) GCN classifiers obey a zero-one law. To achieve our result, we place some mild conditions on the model and initial node embeddings.

First, our study covers sub-Gaussian random vectors, which in particular include all *bounded random vectors*, and all *multivariate normal random vectors*. We note that in every practical setup all node features have bounded values (determined by the bit length of the storage medium), and are thus sub-Gaussian.

**Definition 4.3.** A random vector $\mathbf{x} \in \mathbb{R}^d$ is *sub-Gaussian* if there is $C > 0$ such that for every unit vector $\boldsymbol{y} \in \mathbb{R}^d$ the random variable $\mathbf{x} \cdot \boldsymbol{y}$ satisfies the *sub-Gaussian property*; that is, for every $t > 0$:

$$\mathbb{P}(|\mathbf{x} \cdot \boldsymbol{y}| \geq t) \leq 2 \exp\left(-\frac{t^2}{C^2}\right)$$

Second, we require that the non-linearity $\sigma$ be Lipschitz continuous. This is again a mild assumption, because all non-linearities used in practice are Lipschitz continuous, including $\mathrm{ReLU}$, clipped $\mathrm{ReLU}$, $\mathrm{sigmoid}$, linearized $\mathrm{sigmoid}$ and $\tanh$.

**Definition 4.4.** A function $f\colon \mathbb{R} \to \mathbb{R}$ is *Lipschitz continuous* if there is $C > 0$ such that for any $x, y \in \mathbb{R}$ it holds that $|f(x) - f(y)| \leq C|x - y|$.

Third, we place a condition on the GCN weights with respect to the classifier function $\mathfrak{C}\colon \mathbb{R}^{d(T)} \to \mathbb{B}$, which intuitively excludes a specific weight configuration.

**Definition 4.5.** Consider a distribution $\mathbb{D}(d)$ with mean $\boldsymbol{\mu}$. Let $\mathcal{M}$ be a GCN used for binary graph classification. Define the sequence $\boldsymbol{\mu}_0, \ldots, \boldsymbol{\mu}_T$ of vectors inductively by $\boldsymbol{\mu}_0 := \boldsymbol{\mu}$ and $\boldsymbol{\mu}_t := \sigma(\boldsymbol{W}_n^{(t)} \boldsymbol{\mu}_{t-1} + \boldsymbol{b}^{(t)})$. The classifier classifier $\mathfrak{C} : \mathbb{R}^{d(T)} \to \mathbb{B}$ is *non-splitting* for $\mathcal{M}$ if the vector $\boldsymbol{\mu}_T$ does not lie on a decision boundary for $\mathfrak{C}$.

For all reasonable choices of $\mathfrak{C}$, the decision boundary has dimension lower than the $d(T)$, and is therefore a set of zero-measure. This means that in practice essentially all classifiers are non-splitting.

Given these conditions, we are ready to state our main theorem:

**Theorem 4.6.** *Let $\mathcal{M}$ be a* GCN *used for binary graph classification and take $r \in [0, 1]$. Then, $\mathcal{M}$ satisfies a* zero-one law *with respect to graph distribution $\mathbb{G}(n, r)$ and feature distribution $\mathbb{D}(d)$ assuming the following conditions hold: (i) the distribution $\mathbb{D}(d)$ is sub-Gaussian, (ii) the non-linearity $\sigma$ is Lipschitz continuous, (iii) the graph-level representation uses average pooling, and (iv) the classifier $\mathfrak{C}$ is non-splitting.*

The proof hinges on a probabilistic analysis of the preactivations in each layer. We use a sub-Gaussian concentration inequality to show that the deviation of each of the first-layer preactivations $\mathbf{y}_v^{(1)}$ from its expected value becomes less and less as the number of node $n$ tends to infinity. Using this and the fact that $\sigma$ is Lipschitz continuous we show then that each activation $\mathbf{x}_v^{(1)}$ tends towards a fixed value. Iterating this analysis through all the layers of the network yields the following key lemma, which is the heart of the argument.

**Lemma 4.7.** *Let $\mathcal{M}$ and $\mathbb{D}(d)$ satisfy the conditions in Theorem 4.6. Then, for every layer $t$, there is $\boldsymbol{z}_t \in \mathbb{R}^{d(t)}$ such that when sampling a graph with node features from $\mathbb{G}(n, r)$ and $\mathbb{D}(d)$, for every $i \in \{1, \ldots, d(t)\}$ and for every $\epsilon > 0$ we have that:*

$$\mathbb{P}\left(\forall v \colon \left|\left[\mathbf{x}_v^{(t)} - \boldsymbol{z}_t\right]_i\right| < \epsilon\right) \to 1 \quad as \ n \to \infty$$

With the lemma established, the proof of Theorem 4.6 follows straightforwardly from the last two assumptions. Since the final node embeddings $\mathbf{x}_v^{(T)}$ tend to a fixed value $\boldsymbol{z}_T$, the average-pooled graph-level representations also tend to this. Since we assume that the classifier is non-splitting, this value cannot lie on a decision boundary, and thus the final output is asymptotically stable at $\mathfrak{C}(\boldsymbol{z}_T)$.

We expect that the *rate* of converge will depend in a complex way on the number of layers, the embedding dimensionality, and the choice of non-linearity, which makes a rigorous analysis very challenging. However, considering the manner in which Lemma 4.7 is proved, we can arrive at the

following intuitive argument for why the rate of convergence should decrease as the embedding dimensionality increases: if we fix a node $v$ and a layer $t$ then each of the components of its preactivation can be viewed as the weighted sum of $d(t-1)$ random variables, each of which is the aggregation of activations in the previous layer. Intuitively, as $d(t-1)$ increases, the variance of this sum also increases. This increased variance propagates through the network, resulting in a higher variance for the final node representations and thus a slower convergence.

Using analogous assumptions and techniques to those presented in this section, we also establish a zero-one law for MEANGNN$^+$, which we report in detail in Appendix B. Abstracting away from technicalities, the overall structure of the proofs for MEANGNN$^+$ and GCN is very similar, except that for the former case we additionally need to take care of the global readout component.

## 4.3 Graph neural networks with sum aggregation obey a zero-one law

The other variant of GNNs we consider are those with sum aggregation. The proof in the case works rather differently, and we place different conditions on the model.

**Definition 4.8.** A function $\sigma\colon \mathbb{R} \to \mathbb{R}$ is *eventually constant in both directions* if there are $x_{-\infty}, x_{\infty} \in \mathbb{R}$ such that $\sigma(y)$ is constant for $y < x_{-\infty}$ and $\sigma(y)$ is constant for $y > x_{\infty}$. We write $\sigma_{-\infty}$ to denote the minimum and $\sigma_{\infty}$ to denote the maximum value of an eventually constant function $\sigma$.

This means that there is a threshold ($x_{-\infty}$) below which sigma is constant, and another threshold ($x_{\infty}$) above which sigma is constant. Both the linearized sigmoid and clipped ReLU are eventually constant in both directions. Moreover, when working with finite precision any function with vanishing gradient in both directions (such as sigmoid) can be regarded as eventually constant in both directions.

We also place the following condition on the weights of the model with respect to the mean of $\mathbb{D}(d)$ and the edge-probability $r$.

**Definition 4.9.** Let $\mathcal{M}$ be any SUMGNN$^+$ for binary graph classification with a non-linearity $\sigma$ which is eventually constant in both directions. Let $\mathbb{D}(d)$ be any distribution with mean $\boldsymbol{\mu}$, and let $r \in [0, 1]$. Then, the model $\mathcal{M}$ is *synchronously saturating* for $\mathbb{G}(n, r)$ and $\mathbb{D}(d)$ if the following conditions hold:

1. For each $1 \leq i \leq d(1)$:
$$\left[ (r\boldsymbol{W}_n^{(1)} + \boldsymbol{W}_g^{(1)})\boldsymbol{\mu} \right]_i \neq 0$$

2. For every layer $1 < t \leq T$, for each $1 \leq i \leq d(t)$ and for each $\boldsymbol{z} \in \{\sigma_{-\infty}, \sigma_{\infty}\}^{d(t-1)}$:
$$\left[ (r\boldsymbol{W}_n^{(t)} + \boldsymbol{W}_g^{(t)})\boldsymbol{z} \right]_i \neq 0$$

Analysis of our proof of the zero-one law for SUMGNN$^+$ models (Theorem 4.10 below) reveals that the asymptotic behaviour is determined by the matrices $\boldsymbol{Q}_t := r\boldsymbol{W}_n^{(t)} + \boldsymbol{W}_g^{(t)}$, where the asymptotic final layer embeddings are $\sigma(\boldsymbol{Q}_T(\sigma(\boldsymbol{Q}_{T-1} \cdots \sigma(\boldsymbol{Q}_0\boldsymbol{\mu}) \cdots))))$. To be synchronously saturating is to avoid the boundary case where one of the intermediate steps in the asymptotic final layer embedding computation has a zero component.

Similarly to the case of a non-splitting classifier, the class of synchronously saturating models is very wide. Indeed, the space of models which are not synchronously saturating is the union of solution space of each equality (i.e. the negation of an inequality in Definition 4.9). Thus, assuming that $\boldsymbol{\mu}$, $\sigma_{-\infty}$ and $\sigma_{\infty}$ are non-zero, the space of non-synchronously-saturating models has lower dimension than the space of all models, and thus has measure zero.

With these definitions in place we can now lay out the main result:

**Theorem 4.10.** *Let $\mathcal{M}$ be a* SUMGNN$^+$ *model used for binary graph classification and take $r \in [0, 1]$. Then, $\mathcal{M}$ satisfies a* zero-one law *with respect to graph distribution $\mathbb{G}(n, r)$ and feature distribution $\mathbb{D}(d)$ assuming the following conditions hold: (i) the distribution $\mathbb{D}(d)$ is sub-Gaussian, (ii) the non-linearity $\sigma$ is eventually constant in both directions, (iii) the graph-level representation uses either average or component-wise maximum pooling, and (iv) $\mathcal{M}$ is synchronously saturating for $\mathbb{G}(n, r)$ and $\mathbb{D}(d)$.*

The proof works differently than the GCN and MEANGNN$^+$ cases, but still rests on a probabilistic analysis of the preactivations in each layer. Assuming that $\mathcal{M}$ is synchronously saturating for $\mathbb{G}(n,r)$ and $\mathbb{D}(d)$, we can show that the expected absolute value of each preactivation tends to infinity as the number of nodes increases, and that moreover the probability that it lies below any fixed value tends to 0 exponentially. Hence, the probability that all node embeddings after the first layer are the same and have components which are all $\sigma_{-\infty}$ or $\sigma_{\infty}$ tends to 1. We then extend this analysis to further layers, using the fact that $\mathcal{M}$ is synchronously saturating, which yields inductively that all node embeddings are the same with probability tending to 1, resulting in the following key lemma.

**Lemma 4.11.** *Let $\mathcal{M}$, $\mathbb{D}(d)$ and $r$ be as in Theorem 4.10. Let $\sigma_{-\infty}$ and $\sigma_{\infty}$ be the extremal values taken by the non-linearity. Then, for every layer $t$, there is $\boldsymbol{z}_t \in \{\sigma_{-\infty}, \sigma_{\infty}\}^{d(t)}$ such that when we sample graphs with node features from $\mathbb{G}(n,r)$ and $\mathbb{D}(d)$ the probability that $\mathbf{x}_v^{(t)} = \boldsymbol{z}_t$ for every node $u$ tends to 1 as $n$ tends to infinity.*

The final classification output must therefore be the same asymptotically, since its input consists of node embeddings which always take the same value.

# 5 Graph neural networks with random node features

Up to this point we have been considering the graph plus node features as the (random) input to GNNs. In this section, we make a change in perspective and regard the initial node features as part of the model, so that its input consists solely of the graph without features. We focus in this section on SUMGNN$^+$. Adding random initial features to GNNs is known to increase their power [32].

Note that Theorem 4.10 immediately yields a zero-one law for these models. This places restrictions on what can be expressed by SUMGNN$^+$ models with random features subject to the conditions of Theorem 4.10. For example, it is not possible to express that the number of graph nodes is even, since the property of being even does not satisfy a zero-one law with respect to any $r \in [0,1]$.

It is natural to wonder how tight these restrictions are: what precisely is the class of functions which can be approximated by these models? Let us first formalize the notion of approximation.

**Definition 5.1.** Let $f$ be a Boolean function on graphs, and let $\zeta$ be a random function on graphs. Take $\delta > 0$. Then $\zeta$ *uniformly $\delta$-approximates $f$* if:

$$\forall n \in \mathbb{N}\colon \ \mathbb{P}(\zeta(G) = f(G) \mid |G| = n) \geq 1 - \delta$$

when we sample $G \sim \mathbb{G}(n, 1/2)$.

The reason for sampling graphs from $\mathbb{G}(n, 1/2)$ is that under this distribution all graphs on $n$ nodes are equally likely. Therefore, the requirement is the same as that for every $n \in \mathbb{N}$ the proportion of $n$-node graphs on which $\zeta(G) = f(G)$ is at least $1 - \delta$.

Building on results due to Abboud et al. [1], we show a partial converse to Theorem 4.10: if a graph invariant satisfies a zero-one law for $\mathbb{G}(n, 1/2)$ then it can be universally approximated by a SUMGNN$^+$ with random node features.

**Theorem 5.2.** *Let $\xi$ be any graph invariant which satisfies a zero-one law with respect to $\mathbb{G}(n, 1/2)$. Then, for every $\delta > 0$ there is a SUMGNN$^+$ with random node features $\mathcal{M}$ which uniformly $\delta$-approximates $\xi$.*

The basis of the proof is a result due to Abboud et al. [1] which states that a SUMGNN$^+$ with random node features can approximate any graph invariant *on graphs of bounded size*. When the graph invariant satisfies a zero-one law, we can use the global readout to count the number of nodes. Below a certain threshold, we use the techniques of Abboud et al. [1] to approximate the invariant, and above the threshold we follow its asymptotic behavior. We emphasise that the combination of these techniques yields a model which provides an approximation which is uniform across all graph sizes.

# 6 Experimental evaluation

We empirically verify our theoretical findings on a carefully designed synthetic experiment using ER graphs with random features. The goal of these experiments is to answer the following questions for each model under consideration:

**Q1.** Do we empirically observe a zero-one law?

**Q2.** What is the rate of convergence like empirically?

**Q3.** What is the impact of the number of layers on the convergence?

### 6.1 Experimental setup

We report experiments for GCN, MEANGNN, and SUMGNN. The following setup is carefully designed to eliminate confounding factors:

- We consider 10 GNN models of the same architecture each with *randomly initialized weights*, where each weight is sampled independently from $U(-1, 1)$. The non-linearity is eventually constant in both directions: identity between $[-1, 1]$, and truncated to $-1$ if the input is smaller than $-1$, and $1$ if the input is greater than $1$. In the appendix we include additional experiments in which test other choices of non-linearity (see Appendix E.2). We apply mean pooling to yield a final representation $\mathbf{z}_G \in \mathbb{R}^d$ of the input graph.

- For every model, we apply a final classifier $\sigma(f) : \mathbb{R}^d \to \mathbb{B}$ where $f$ is a 2-layer MLP with *random weights* and with $\tanh$ activation, which outputs a real value, and $\sigma$ is the sigmoid function. Graphs are classified as $1$ if the output of the sigmoid is greater than $0.5$, and $0$ otherwise.

- The input graphs are drawn from $\mathbb{G}(n, 1/2)$ with corresponding node features independently drawn from $U(0, 1)$.

- We conduct these experiments with three choices of layers: 10 models with $T = 1$ layer, 10 models with $T = 2$ layers, and 10 models with $T = 3$ layers.

The goal of these experiments is to understand the behavior of the respective GNN graph classifiers with mean-pooling, as we draw larger and larger ER graphs. Specifically, each model classifies graphs of varying sizes, and we are interested in knowing *how the proportion of the graphs which are classified as $1$ evolves, as we increase the graph sizes*.

We independently sample 10 models to ensure this is not a model-specific behavior, aiming to observe the same phenomenon across the models. If there is a zero-one law, then for each model, we should only see two types of curves: either tending to $0$ or tending to $1$, as graph sizes increase. Whether it will tend to $0$ or $1$ depends on the final classifier: since each of these are independent MLPs with random weights the specific outcome is essentially random.

We consider models with up to 3 layers to ensure that the node features do not become alike because of the orthogonal over-smoothing issue [24], which surfaces with increasing number of layers. A key feature of our theoretical results is that they do not depend on the number of layers, and this is an aspect which we wish to validate empirically. Using models with random weights is a neutral setup, and random GNNs are widely used in the literature as baseline models [36], as they define valid graph convolutions and tend to perform reasonably well.

### 6.2 Empirical results

We report all results in Figure 1 for all models considered and discuss them below. Each plot in this figure depicts the curves corresponding to the behavior of independent models with random weights.

**GCN.** For this experiment, we use an embedding dimensionality of 128 for each GCN model and draw graphs of sizes up 5000, where we take 32 samples of each size. The key insight of Theorem 4.6 is that the final mean-pooled embedding vector $\mathbf{z}_G$ tends to a constant vector as we draw larger graphs. Applying an MLP followed by a sigmoid function will therefore map $\mathbf{z}_G$ to either $0$ or $1$, showing a zero-one law. It is evident from Figure 1 (top row) that all curves tend to either $0$ or $1$, confirming our expectation regarding the outcome of these experiments for GCNs. Moreover, this holds regardless of the number of layers considered. Since the convergence occurs quickly, already around graphs of size of 1000, we did not experiment with larger graph sizes in this experiment.

**MEANGNN.** Given that the key insight behind this result is essentially similar to that of Theorem 4.6, we follow the exact same configuration for these models as for GCNs. The proof structure is the same in both cases: we show that the preactivations and activations become closer and closer to

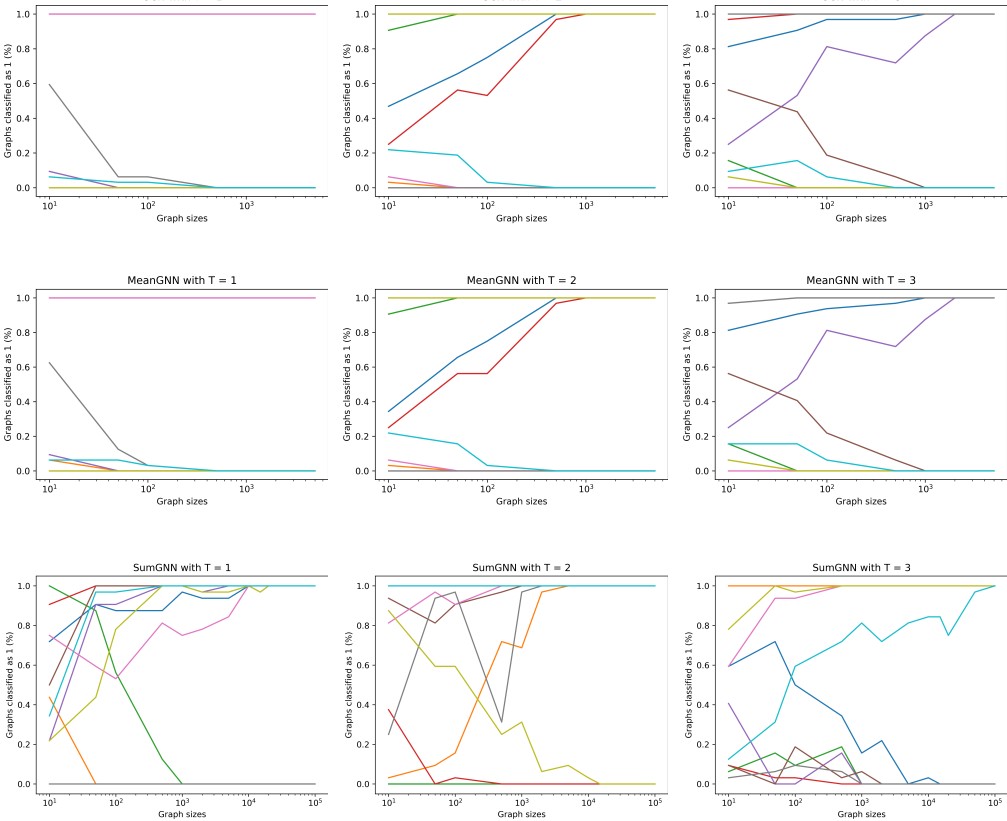

Figure 1: Each plot shows the proportion of graphs of certain size which are classified as $1$ by a set of ten GCNs (top row), MEANGNNs (middle), and SUMGNNs (bottom row). Each curve (color-coded) shows the behavior of a model, as we draw increasingly larger graphs. The phenomenon is observed for 1-layer models (left column), 2-layer models (mid column), and 3-layer models (last column). GCNs and MEANGNNs behave very similarly with all models converging quickly to $0$ or to $1$. SUMGNNs shows slightly slower convergence, but all models perfectly converge in all layers.

some fixed values as the number of nodes increases. Moreover, comparing the summations in the definitions of GCN and MEANGNN, on a typical ER graph drawn from $\mathbb{G}(n, 1/2)$ we would expect each corresponding summand to have a similar value, since $\sqrt{|\mathcal{N}(u)||\mathcal{N}(v)|}$ should be close to $|\mathcal{N}^+(v)|$. Figure 1(mid row) illustrates the results for MEANGNN and the trends are reassuringly similar to those of GCNs: all models converge quickly to either $0$ and $1$ with all choices of layers. Interestingly, the plots for GCN and MEANGNN models are almost identical. We used the same seed when drawing each of the model weights, and the number of parameters is the same between the two. Hence, the GCN models were parameterized with the same values as the MEANGNN models. The fact that each pair of models preforms near identically confirms the expectation that the two architectures work in similar ways on ER graphs.

**SUMGNN.** Theorem 4.10 shows that, as the number of nodes grow, the embedding vector $\mathbf{z}_v$ of each *node* $v$ will converge to a constant vector with high probability. Hence, when we do mean-pooling at the end, we expect to get the same vector for different graphs of the same size. The mechanism by which a zero-one law is arrived at is quite different compared with the GCN and MEANGNN case. In particular, in order for the embedding vectors to begin to converge, there must be sufficiently many nodes so that the preactivations surpass the thresholds of the non-linearity. For this experiment, we use a smaller embedding dimensionality of $64$ for each SUMGNN model and draw graphs of sizes up to $100000$, where we take 32 samples of each size. Figure 1 shows the results for SUMGNN. Note that we observe a slower convergence than with GCN or MEANGNN.

# 7 Limitations, discussion, and outlook

The principal limitations of our work come from the assumptions placed on the main theorems. In our formal analysis, we focus on graphs drawn from the ER distribution. From the perspective of characterizing the *expressiveness* of GNNs this is unproblematic, and accords with the classical analysis of first-order properties of graphs. However, when considering the *extrapolation capacity* of GNNs, other choices of distributions may be more realistic. In Appendices E.4 and E.5 we report experiments in which zero-one laws are observed empirically for sparse ER graphs and Barabási-Albert graphs [3], suggesting that formal results may be obtainable. While we empirically observe that GNNs converge to their asymptotic behaviour very quickly, we leave it as future work to rigorously examine the rate at which this convergence occurs.

In this work we show that GNNs with random features can at most capture properties which follow a zero-one law. We complement this with an *almost* matching lower bound: Theorem 5.2 currently requires a graph invariant $\xi$ which obeys a zero-one law with respect to a specific value of $r$ (i.e., $1/2$), and if this assumption could be relaxed, it would yield a complete characterization of the expressive power of these models.

# 8 Acknowledgments

We would like to thank the anonymous reviewers for their feedback, which lead to several improvements in the presentation of the paper. The first author was supported by an EPSRC studentship with project reference *2271793*.

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

# A   Proof of the zero-one law for GCN

The proof of Lemma 4.7 and subsequently Theorem 4.6 relies on an asymptotic analysis of the distributions of the node embeddings at each layer. The following famous concentration inequality for sub-Gaussian random variables allows us to put bounds on the deviation of a sum of random variables from its expected value.

**Theorem A.1** (Hoeffding Inequality for sub-Gaussian random variables)**.** *There is a universal constant $c$ such that the following holds. Let $z_1, \dots, z_N$ be independent sub-Gaussian scalar random variables with mean $0$. Assume that the constants $C$ from Definition 4.3 for each $z_i$ can be bounded by $K$. Then for all $t > 0$:*

$$\mathbb{P}\left(\left|\sum_{i=1}^{N} z_i\right| \geq t\right) \leq \exp\left(-\frac{ct^2}{K^2 N}\right)$$

*Proof.* See Theorem 2.6.2 in [38]. $\qquad\square$

We also make use of the following three basic facts about sub-Gaussian random variables.

**Lemma A.2.** *If $z$ is a sub-Gaussian random vector and $q$ is any vector of the same dimension then $q \cdot z$ is sub-Gaussian.*

*Proof.* This follows directly from Definition 4.3. $\qquad\square$

**Lemma A.3.** *If $z$ is a sub-Gaussian scalar random variable then so is $z - \mathbb{E}[z]$.*

*Proof.* See Lemma 2.6.8 in [38]. $\qquad\square$

**Lemma A.4.** *If $z$ is a sub-Gaussian scalar random variable and $a$ is an independent Bernoulli random variable then $za$ is sub-Gaussian.*

*Proof.* Let $C$ be the constant given by Definition 4.3 for $z$. Let $a$ take values $\alpha$ and $\beta$. Using the Law of Total Probability:

$$\begin{aligned}
\mathbb{P}(|za| \geq t) &= \mathbb{P}(|za| \geq t \mid a = \alpha)\,\mathbb{P}(a = \alpha) + \mathbb{P}(|za| \geq t \mid a = \beta)\,\mathbb{P}(a = \beta) \\
&= \mathbb{P}(|z| \geq t/|\alpha|)\,\mathbb{P}(a = \alpha) + \mathbb{P}(|z| \geq t/|\beta|)\,\mathbb{P}(a = \beta) \\
&\leq 2\exp\left(-\frac{t^2}{|\alpha|^2 C^2}\right)\mathbb{P}(a = \alpha) + 2\exp\left(-\frac{t^2}{|\beta|^2 C^2}\right)\mathbb{P}(a = \beta) \\
&\leq 2\exp\left(-\frac{t^2}{\max\{|\alpha|, |\beta|\}^2 C^2}\right)
\end{aligned}$$

Therefore $za$ is sub-Gaussian. $\qquad\square$

We first prove the key lemma regarding the node embeddings.

*Proof of Lemma 4.7.* Let $C$ be the Lipschitz constant for $\sigma$. Start by considering the first layer preactivations $\mathbf{y}_v^{(1)}$ and drop superscript $(1)$'s for notational clarity. We have that:

$$\mathbf{y}_v = \sum_{v \in \mathcal{N}^+(v)} \frac{1}{\sqrt{|\mathcal{N}(v)||\mathcal{N}(u)|}} \boldsymbol{W}_n \mathbf{x}_u^{(0)} + \boldsymbol{b}$$

Fix $i \in \{1, \dots, d(1)\}$. The deviation from the expected value in the $i$th component is as follows:

$$|[\mathbf{y}_v - \mathbb{E}[\mathbf{y}_v]]_i| = \left|\sum_{u \in \mathcal{N}^+(v)} \frac{1}{\sqrt{|\mathcal{N}(v)||\mathcal{N}(u)|}}\left[\boldsymbol{W}_n \mathbf{x}_u^{(0)} - \mathbb{E}\left[\boldsymbol{W}_n \mathbf{x}_u^{(0)}\right]\right]_i\right|$$

Now every $|\mathcal{N}(u)|$ is a sum of $n$ independent 0-1 Bernoulli random variables with success probability $r$ (since the graph is sampled from an Erdős-Rényi distribution). Since Bernoulli random variables are sub-Gaussian (Lemma A.4) we can use Hoeffding's Inequality to bound the deviation of $|\mathcal{N}^+(u)| =$

$|\mathcal{N}(u)| + 1$ from its expected value $nr + 1$. By Theorem A.1 there is a constant $K$ such that for every $\gamma \in (0,1)$ and node $u$:

$$\mathbb{P}(|\mathcal{N}^+(u)| \leq \gamma nr) \leq \mathbb{P}(||\mathcal{N}^+(u)| - nr| \geq (1-\gamma)nr)$$
$$\leq \mathbb{P}(||\mathcal{N}(u)| - nr| \geq (1-\gamma)nr - 1)$$
$$\leq 2\exp\left(-\frac{K((1-\gamma)nr - 1)^2}{n}\right)$$

This means that, taking a union bound:

$$\mathbb{P}(\forall u \in V : |\mathcal{N}^+(u)| \geq \gamma nr) \geq 1 - 2n\exp\left(-\frac{K_0((1-\gamma)nr - 1)^2}{n}\right)$$

Fix $i \in \{1, \ldots, d(1)\}$. In the case where $\forall u \in V : |\mathcal{N}^+(u)| \geq \gamma nr$ we have that:

$$|[\mathbf{y}_v - \mathbb{E}[\mathbf{y}_v]]_i| \leq \frac{1}{\sqrt{|\mathcal{N}(v)|\gamma nr}} \left|\sum_{u \in \mathcal{N}^+(v)} \left[\boldsymbol{W}_n \mathbf{x}_u^{(0)} - \mathbb{E}\left[\boldsymbol{W}_n \mathbf{x}_u^{(0)}\right]\right]_i\right|$$

Now, by Lemma A.2 and Lemma A.3 each $\boldsymbol{W}_n \mathbf{x}_u^{(0)} - \mathbb{E}\left[\boldsymbol{W}_n \mathbf{x}_u^{(0)}\right]$ is sub-Gaussian. We can thus apply Hoeffding's Inequality (Theorem A.1) to obtain a constant $K$ such that for every $t > 0$ we have:

$$\mathbb{P}(|[\mathbf{y}_v - \mathbb{E}[\mathbf{y}_v]]_i| \geq t) \leq \mathbb{P}\left(\left|\sum_{u \in \mathcal{N}^+(v)} \left[\boldsymbol{W}_n \mathbf{x}_u^{(0)} - \mathbb{E}\left[\boldsymbol{W}_n \mathbf{x}_u^{(0)}\right]\right]_i\right| \geq t\sqrt{|\mathcal{N}(v)|\gamma nr}\right)$$
$$\leq 2\exp\left(-\frac{Kt^2|\mathcal{N}(v)|\gamma nr}{|\mathcal{N}^+(v)|}\right)$$
$$\leq 2\exp\left(-\frac{Kt^2\gamma nr}{2}\right)$$

Now using the Law of Total Probability, partitioning depending on whether $\forall u \in V : |\mathcal{N}^+(u)| \geq \gamma nr$, we get a bound as follows:

$$\mathbb{P}(|[\mathbf{y}_v - \mathbb{E}[\mathbf{y}_v]]_i| \geq t) \leq 2\exp\left(-\frac{Kt^2\gamma nr}{2}\right) + 2n\exp\left(-\frac{K_0((1-\gamma)nr - 1)^2}{n}\right)$$

From now on fix any $\gamma \in (0,1)$.

Let $\mathbf{z}_1 := \sigma(\mathbb{E}[\mathbf{y}_v])$ for any $v$ (this is the same for every $v$). Applying the bound with $t = C\epsilon$ we can bound the deviation of $\mathbf{x}_v$ from $\mathbf{z}_1$ as follows, using the Lipschitz constant $C$.

$$\mathbb{P}(|[\mathbf{x}_v - \mathbf{z}_1]_i| \geq \epsilon) = \mathbb{P}(|[\sigma(\mathbf{y}_v) - \sigma(\mathbb{E}[\mathbf{y}_v])]_i| \geq \epsilon)$$
$$\leq \mathbb{P}(|[\mathbf{y}_v - \mathbb{E}[\mathbf{y}_v]]_i| \geq C\epsilon)$$
$$\leq 2\exp\left(-\frac{KC^2\epsilon^2\gamma nr}{2}\right) + 2n\exp\left(-\frac{K_0((1-\gamma)nr - 1)^2}{n}\right)$$

Taking a union bound, the probability that $|[\mathbf{x}_v - \mathbf{z}_1]_i| < \epsilon$ for every node $v$ and every $i \in \{1, \ldots, d(1)\}$ is at least:

$$1 - nd(i)\,\mathbb{P}(|[\mathbf{x}_v - \mathbf{z}_1]_i| \geq \epsilon)$$

This tends to $1$ as $n$ tends to infinity, which yields the result for the first layer.

Now consider the preactivations for the second layer:

$$\mathbf{y}_v^{(2)} = \sum_{u \in \mathcal{N}^+(v)} \frac{1}{\sqrt{|\mathcal{N}(v)||\mathcal{N}(u)|}} \boldsymbol{W}_n^{(2)} \mathbf{x}_u^{(1)} + \boldsymbol{b}^{(2)}$$

As in the single layer case, we can bound the probability that any $|\mathcal{N}(u)|$ is less than some $\gamma nr$. Condition on the event that $\forall u \in V : |\mathcal{N}^+(u)| \geq \gamma nr$.

By applying the result for the first layer to $\epsilon' = \epsilon\sqrt{\gamma r}/(2C\|\boldsymbol{W}_n^{(2)}\|_\infty)$, we have that for each $i \in \{1, \ldots, d(2)\}$:

$$\mathbb{P}\left(\forall v\colon \left|\left[\mathbf{x}_v^{(1)} - \boldsymbol{z}_1\right]_i\right| < \epsilon'\right) \to 1 \quad \text{as } n \to \infty$$

Condition additionally on the event that $|[\mathbf{x}_v^{(1)} - \boldsymbol{z}_1]_i| < \epsilon'$ for every node $v$ and every $i \in \{1, \ldots, d(1)\}$.

Now define:

$$\boldsymbol{a}_2 := \sum_{v \in \mathcal{N}^+(v)} \frac{1}{\sqrt{|\mathcal{N}(v)||\mathcal{N}(u)|}} \boldsymbol{W}_n^{(2)} \boldsymbol{z}_1 + \boldsymbol{b}^{(2)}$$

Then we have that for every $i \in \{1, \ldots, d(1)\}$:

$$
\begin{aligned}
|[\mathbf{y}_v^{(2)} - \boldsymbol{a}_2]_i| &\leq \left|\sum_{u \in \mathcal{N}^+(v)} \frac{1}{\sqrt{|\mathcal{N}(u)||\mathcal{N}(v)|}} \left[\boldsymbol{W}_n^{(2)}(\mathbf{x}_u^{(1)} - \boldsymbol{z}_1)\right]_i\right| \\
&\leq \frac{1}{\sqrt{|\mathcal{N}(v)|\gamma n r}} \left|\sum_{u \in \mathcal{N}^+(v)} \left[\boldsymbol{W}_n^{(2)}(\mathbf{x}_u^{(1)} - \boldsymbol{z}_1)\right]_i\right| \\
&\leq \frac{1}{\sqrt{|\mathcal{N}(v)|\gamma n r}} \left\|\boldsymbol{W}_n^{(2)}\right\|_\infty \sum_{u \in \mathcal{N}^+(v)} \left\|\mathbf{x}_u^{(1)} - \boldsymbol{z}_1\right\|_\infty \\
&\leq \frac{\epsilon|\mathcal{N}^+(v)|}{2C\sqrt{|\mathcal{N}(v)|n}} \\
&\leq \frac{\epsilon(n+1)}{2Cn} \\
&\leq \frac{\epsilon}{C}
\end{aligned}
$$

Now let $\boldsymbol{z}_2 := \sigma(\boldsymbol{a}_2)$. As in the single-layer case we can use the bound on $|[\mathbf{y}_v^{(2)} - \boldsymbol{a}_2]_i|$ and the fact that $\sigma$ is Lipschitz to find that, for every node $v$ and $i \in \{1, \ldots, d(2)\}$:

$$|[\mathbf{x}_v^{(2)} - \boldsymbol{z}_2]_i| < \epsilon$$

Since the probability that the two events on which we conditioned tends to 1, the result follows for the second layer.

Finally, we apply the argument inductively through the layers to obtain the ultimate result. $\qquad\square$

With the key lemma established we can prove the main result.

*Proof of Theorem 4.6.* By Lemma 4.7 the final node embeddings $\mathbf{x}_v^{(T)}$ deviate less and less from $\boldsymbol{z}_T$ as the number of nodes $n$ increases. Therefore, the average-pooled graph-level representation also deviates less and less from $\boldsymbol{z}_T$. By inspecting the proof, we can see that this $\boldsymbol{z}_T$ is exactly the vector $\boldsymbol{\mu}_T$ in the definition of non-splitting (Definition 4.5). This means that $\boldsymbol{z}_T$ cannot lie on a decision boundary for the classifier $\mathfrak{C}$. Hence, there is $\epsilon > 0$ such that $\mathfrak{C}$ is constant on:

$$\{\boldsymbol{x} \in \mathbb{R}^{d(T)} \mid \forall i \in \{1, \ldots, d(T)\}\colon [\boldsymbol{z}_T - \boldsymbol{x}]_i < \epsilon\}$$

Therefore, the probability that the output of $\mathcal{M}$ is $\mathfrak{C}(\boldsymbol{z}_T)$ tends to 1 as $n$ tends to infinity. $\qquad\square$

# B   Proof of the zero-one law for MEANGNN$^+$

Let us turn now to establishing a zero-one law for GNNs using mean aggregation. We place the same conditions as with Theorem 4.6. This time the notion of 'non-splitting' is as follows.

**Definition B.1.** Consider a distribution $\mathbb{D}(d)$ with mean $\boldsymbol{\mu}$. Let $\mathcal{M}$ be a MEANGNN$^+$ used for binary graph classification. Define the sequence $\boldsymbol{\mu}_0, \ldots, \boldsymbol{\mu}_T$ of vectors inductively by $\boldsymbol{\mu}_0 := \boldsymbol{\mu}$ and $\boldsymbol{\mu}_t := \sigma((\boldsymbol{W}_n^{(t)} + \boldsymbol{W}_r^{(t)})\boldsymbol{\mu}_{t-1} + \boldsymbol{b}^{(t)})$. The classifier $\mathfrak{C}\colon \mathbb{R}^{d(T)} \to \mathbb{B}$ is *non-splitting* for $\mathcal{M}$ if the vector $\boldsymbol{\mu}_T$ does not lie on a decision boundary for $\mathfrak{C}$.

Again, in practice essentially all classifiers are non-splitting.

**Theorem B.2.** *Let $\mathcal{M}$ be a* MEANGNN$^+$ *used for binary graph classification and take $r \in [0,1]$. Then, $\mathcal{M}$ satisfies a* zero-one law *with respect to graph distribution $\mathbb{G}(n,r)$ and feature distribution $\mathbb{D}(d)$ assuming the following conditions hold: (i) the distribution $\mathbb{D}(d)$ is sub-Gaussian, (ii) the non-linearity $\sigma$ is Lipschitz continuous, (iii) the graph-level representation uses average pooling, (iv) the classifier $\mathfrak{C}$ is non-splitting.*

Note that the result immediately applies to the MEANGNN architecture, since it is a special case of MEANGNN$^+$.

The overall structure of the proof is the same as for GCN. In particular, we prove the following key lemma stating that all node embeddings tend to fixed values.

**Lemma B.3.** *Let $\mathcal{M}$ and $\mathbb{D}(d)$ satisfy the conditions in Theorem B.2. Then, for every layer $t$, there is $\boldsymbol{z}_t \in \mathbb{R}^{d(t)}$ such when sampling a graph with node features from $\mathbb{G}(n,r)$ and $\mathbb{D}(d)$, for every $i \in \{1, \ldots, d(t)\}$ and for every $\epsilon > 0$ we have that:*

$$\mathbb{P}\left(\forall v \colon \left|\left[\mathbf{x}_v^{(t)} - \boldsymbol{z}_t\right]_i\right| < \epsilon\right) \to 1 \quad as\ n \to \infty$$

We are ready to present the proofs of the statements. The proof of the key lemma works in a similar way to the GCN case.

*Proof of Lemma B.3.* Let $C$ be the Lipschitz constant for $\sigma$. Start by considering the first layer preactivations $\mathbf{y}_v^{(1)}$ and drop superscript $(1)$'s for notational for clarity. We have that:

$$\mathbf{y}_v = \frac{1}{|\mathcal{N}^+(v)|} \boldsymbol{W}_n \sum_{v \in \mathcal{N}^+(u)} \mathbf{x}_v^{(0)} + \frac{1}{n} \boldsymbol{W}_r \sum_{u \in V} \mathbf{x}_u^{(0)} + \boldsymbol{b}$$

Fix $i \in \{1, \ldots, d(1)\}$. We can bound the deviation from the expected value as follows:

$$\left|[\mathbf{y}_v - \mathbb{E}[\mathbf{y}_v]]_i\right| \leq \frac{1}{|\mathcal{N}^+(v)|} \left|\sum_{u \in \mathcal{N}^+(v)} \left[\boldsymbol{W}_n \mathbf{x}_u^{(0)} - \mathbb{E}\left[\boldsymbol{W}_n \mathbf{x}_u^{(0)}\right]\right]_i\right| + \frac{1}{n}\left|\sum_{u \in V} \left[\boldsymbol{W}_n \mathbf{x}_u^{(0)} - \mathbb{E}\left[\boldsymbol{W}_n \mathbf{x}_u^{(0)}\right]\right]_i\right|$$

By Lemma A.2 and Lemma A.3 both each $\left[\boldsymbol{W}_n \mathbf{x}_u^{(0)} - \mathbb{E}\left[\boldsymbol{W}_n \mathbf{x}_u^{(0)}\right]\right]_i$ and each $\left[\boldsymbol{W}_n \mathbf{x}_u^{(0)} - \mathbb{E}\left[\boldsymbol{W}_n \mathbf{x}_u^{(0)}\right]\right]_i$ are sub-Gaussian. We can therefore apply Hoeffding's Inequality to their sums. First, by Theorem A.1 there is a constant $K_{\mathrm{g}}$ such that for any $t > 0$:

$$\mathbb{P}\left(\frac{1}{n}\left|\sum_{u \in V}\left[\boldsymbol{W}_n\mathbf{x}_u^{(0)} - \mathbb{E}\left[\boldsymbol{W}_n\mathbf{x}_u^{(0)}\right]\right]_i\right| \leq t\right) = \mathbb{P}\left(\left|\sum_{u \in V}\left[\boldsymbol{W}_n\mathbf{x}_u^{(0)} - \mathbb{E}\left[\boldsymbol{W}_n\mathbf{x}_u^{(0)}\right]\right]_i\right| \leq tn\right)$$

$$\leq 2\exp\left(-\frac{K_{\mathrm{g}}t^2 n^2}{n}\right)$$

$$= 2\exp\left(-K_{\mathrm{g}}t^2 n\right)$$

Second, applying Theorem A.1 again there is a constant $K_{\mathrm{n}}$ such that for any $t > 0$:

$$\mathbb{P}\left(\frac{1}{|\mathcal{N}^+(v)|}\left|\sum_{u \in \mathcal{N}^+(v)}\left[\boldsymbol{W}_n\mathbf{x}_u^{(0)} - \mathbb{E}\left[\boldsymbol{W}_n\mathbf{x}_u^{(0)}\right]\right]_i\right| \leq t\right)$$

$$= \mathbb{P}\left(\left|\sum_{u \in \mathcal{N}^+(v)}\left[\boldsymbol{W}_n\mathbf{x}_u^{(0)} - \mathbb{E}\left[\boldsymbol{W}_n\mathbf{x}_u^{(0)}\right]\right]_i\right| \leq t|\mathcal{N}^+(v)|\right)$$

$$\leq 2\exp\left(-\frac{K_{\mathrm{n}}t^2|\mathcal{N}^+(v)|^2}{n}\right)$$

Now $|\mathcal{N}(v)|$ is a the sum of $n$ independent 0-1 Bernoulli random variables with success probability $r$. Hence, as in the proof of Lemma 4.7, by Hoeffding's Inequality (Theorem A.1) there is a constant $K_0$ such that for every $\gamma \in (0,1)$:

$$\mathbb{P}(|\mathcal{N}^+(v)| \geq \gamma n r) \leq 2 \exp\left(-\frac{K_0((1-\gamma)nr - 1)^2}{n}\right)$$

We can then use the Law of Total Probability, partitioning on whether $|\mathcal{N}^+(v)| \geq \gamma n r$, to get a bound as follows:

$$\mathbb{P}\left(\frac{1}{|\mathcal{N}^+(v)|}\left|\sum_{u \in \mathcal{N}^+(v)} \left(\boldsymbol{W}_n \mathbf{x}_u^{(0)} - \mathbb{E}\left[\boldsymbol{W}_n \mathbf{x}_u^{(0)}\right]\right)\right| \leq t\right)$$

$$\leq 2 \exp\left(-\frac{K_\mathrm{n} t^2 (\gamma n r)^2}{n}\right) + 2 \exp\left(-\frac{K_0((1-\gamma)nr - 1)^2}{n}\right)$$

From now on fix any $\gamma \in (0,1)$.

Finally let $\boldsymbol{z}_1 := \sigma(\mathbb{E}[\mathbf{y}_v])$ for any $v$ (this is the same for every $v$). Applying the two bounds with $t = C\epsilon/2$ we can bound the deviation of $\mathbf{x}_v$ from $\boldsymbol{z}_1$ as follows, using the Lipschitz constant $C$.

$$\mathbb{P}(|[\mathbf{x}_v - \boldsymbol{z}_1]_i| \geq \epsilon) = \mathbb{P}(|[\sigma(\mathbf{y}_v) - \sigma(\mathbb{E}[\mathbf{y}_v])]_i| \geq \epsilon)$$
$$\leq \mathbb{P}(|[\mathbf{y}_v - \mathbb{E}[\mathbf{y}_v]]_i| \geq C\epsilon)$$
$$\leq \left(\begin{array}{c} \mathbb{P}\left(\frac{1}{|\mathcal{N}^+(v)|}\left|\sum_{u \in \mathcal{N}^+(v)}\left[\boldsymbol{W}_n \mathbf{x}_u^{(0)} - \mathbb{E}\left[\boldsymbol{W}_n \mathbf{x}_u^{(0)}\right]\right]\right|_i \leq \frac{C\epsilon}{2}\right) \\ + \mathbb{P}\left(\frac{1}{n}\left|\sum_{u \in V}\left[\boldsymbol{W}_n \mathbf{x}_u^{(0)} - \mathbb{E}\left[\boldsymbol{W}_n \mathbf{x}_u^{(0)}\right]\right]\right|_i \leq \frac{C\epsilon}{2}\right) \end{array}\right)$$
$$\leq \left(\begin{array}{c} 2\exp\left(-\frac{K_\mathrm{n}(C\epsilon\gamma r)^2 n}{4}\right) \\ +2\exp\left(-\frac{K_0((1-\gamma)nr-1)^2}{n}\right) \\ +2\exp\left(-K_\mathrm{g}\frac{(C\epsilon)^2 n}{4}\right) \end{array}\right)$$

Taking a union bound, the probability that $|[\mathbf{x}_v - \boldsymbol{z}_1]_i| < \epsilon$ for every node $v$ and every $i \in \{1, \ldots, d(1)\}$ is at least:

$$1 - n d(i)\,\mathbb{P}(|[\mathbf{x}_v - \boldsymbol{z}_1]_i| \geq \epsilon)$$

This tends to 1 as $n$ tends to infinity.

Let us turn now to the second layer. By applying the above result for the first layer to $\epsilon' = \epsilon/(C \max\{\|\boldsymbol{W}_n^{(2)}\|_\infty), \|\boldsymbol{W}_r^{(2)}\|_\infty\}$, we have that for each $i \in \{1, \ldots, d(2)\}$:

$$\mathbb{P}\left(\forall v : \left|\left[\mathbf{x}_v^{(1)} - \boldsymbol{z}_1\right]_i\right| < \epsilon'\right) \to 1 \quad \text{as } n \to \infty$$

Condition on the event that $|[\mathbf{x}_v^{(1)} - \boldsymbol{z}_1]_i| < \epsilon'$ for every node $v$ and every $i \in \{1, \ldots, d(1)\}$.

Fix $v$ and consider the second-layer preactivations:

$$\mathbf{y}_v^{(2)} = \frac{1}{|\mathcal{N}^+(v)|}\boldsymbol{W}_n^{(2)}\sum_{u \in \mathcal{N}^+(v)}\mathbf{x}_u^{(1)} + \frac{1}{n}\boldsymbol{W}_r^{(2)}\sum_{u \in V}\mathbf{x}_u^{(1)} + \boldsymbol{b}^{(2)}$$

Define:

$$\boldsymbol{a}_2 := \frac{1}{|\mathcal{N}^+(v)|}\boldsymbol{W}_n^{(2)}\sum_{u \in \mathcal{N}^+(v)}\boldsymbol{z}_1 + \frac{1}{n}\boldsymbol{W}_r^{(2)}\sum_{u \in V}\boldsymbol{z}_1 + \boldsymbol{b}^{(2)}$$

Fix $i \in \{1, \ldots, d(2)\}$. Then:

$$\left|[\mathbf{y}_v^{(2)} - \boldsymbol{a}_2]_i\right| = \left|\frac{1}{|\mathcal{N}^+(v)|}\boldsymbol{W}_n^{(2)}\sum_{u \in \mathcal{N}^+(v)}(\mathbf{x}_u^{(1)} - \boldsymbol{z}_1) + \frac{1}{n}\boldsymbol{W}_r^{(2)}\sum_{u \in V}(\mathbf{x}_u^{(1)} - \boldsymbol{z}_1)\right|$$
$$\leq \frac{1}{|\mathcal{N}^+(v)|}\left\|\boldsymbol{W}_n^{(2)}\right\|_\infty\sum_{u \in \mathcal{N}^+(v)}\left\|\mathbf{x}_u^{(1)} - \boldsymbol{z}_1\right\|_\infty + \frac{1}{n}\left\|\boldsymbol{W}_r^{(2)}\right\|_\infty\sum_{u \in V}\left\|\mathbf{x}_u^{(1)} - \boldsymbol{z}_1\right\|_\infty$$
$$\leq \frac{\epsilon}{C}$$

Let $z_2 := \sigma(a_2)$. Then we can use the Lipschitz continuity of $\sigma$ to bound the deviation of the activation from $z_2$ as follows.

$$\left| [\mathbf{x}_v^{(2)} - z_2]_i \right| = \left| [\sigma(\mathbf{y}_v^{(2)}) - \sigma(a_2)]_i \right| \leq C \left| [\mathbf{y}_v^{(2)} - a_2]_i \right| \leq \epsilon$$

Since the probability that $|[\mathbf{x}_v^{(1)} - z_1]_i| < \epsilon'$ for every node $v$ and every $i \in \{1, \ldots, d(1)\}$ tends to 1, we get that the probability that $|[\mathbf{x}_v^{(2)} - z_2]_i| < \epsilon$ for every node $v$ and every $i \in \{1, \ldots, d(2)\}$ also tends to 1.

Finally we apply the above argument inductively through all layers to get the desired result. $\square$

The proof of the main result now proceeds as in the proof of Theorem 4.6.

*Proof of Theorem B.2.* By Lemma B.3 the final node embeddings $\mathbf{x}_v^{(T)}$ deviate less and less from $z_T$ as the number of nodes $n$ increases. Therefore, the average-pooled graph-level representation also deviates less an less from $z_T$. By inspecting the proof, we can see that this $z_T$ is exactly the vector $\mu_T$ in the definition of non-splitting (Definition B.1). This means that $z_T$ cannot lie on a decision boundary for the classifier $\mathfrak{C}$. Hence, there is $\epsilon > 0$ such that $\mathfrak{C}$ is constant on:

$$\{x \in \mathbb{R}^{d(T)} \mid \forall i \in \{1, \ldots, d(T)\} \colon [z_T - x]_i < \epsilon\}$$

Therefore, the probability that the output of $\mathcal{M}$ is $\mathfrak{C}(z_T)$ tends to 1 as $n$ tends to infinity. $\square$

## C   Proof of the zero-one law for SUMGNN⁺

The proof of the key lemma works rather differently to the GCN and MEANGNN⁺ case, but we still make important use of Hoeffding's Inequality.

*Proof of Lemma 4.11.* Consider the first layer preactivations $\mathbf{y}_v^{(1)}$ and drop superscript (1)'s for notational clarity. We can rearrange the expression as follows:

$$\mathbf{y}_v = (\boldsymbol{W}_s + \boldsymbol{W}_g)\mathbf{x}_v^{(0)} + (\boldsymbol{W}_n + \boldsymbol{W}_g) \sum_{u \in \mathcal{N}(v)} \mathbf{x}_u^{(0)} + \boldsymbol{W}_g \sum_{u \in V \setminus \mathcal{N}^+(v)} \mathbf{x}_u^{(0)} + \boldsymbol{b}$$

For $u, v \leq n$ define:

$$\mathbf{w}_{u,v} = (\mathrm{A}_{uv}\boldsymbol{W}_n + \boldsymbol{W}_g)\mathbf{x}_u^{(0)} 1_{u \neq v} + (\boldsymbol{W}_s + \boldsymbol{W}_g)\mathbf{x}_u^{(0)} 1_{u=v}$$

Using this, we can rewrite:

$$\mathbf{y}_v = \sum_{u=1}^{n} \mathbf{w}_{u,v} + \boldsymbol{b}$$

By assumption on the distribution from which we draw graphs with node features, the $\mathbf{w}_{u,v}$'s are independent for any fixed $v$.

Now fix $i \in \{1, \ldots, d(1)\}$. By Lemma A.2 and Lemma A.4 we have that each $[\mathbf{w}_{u,v}]_i$ is sub-Gaussian. We therefore apply Hoeffding's Inequality to the sum. Note that $\mathbf{w}_{u,v}$ can have one of two (sub-Gaussian) distributions, depending on whether $u = v$. Therefore, by Theorem A.1 and Lemma A.3, there are constants $c$ and $K$ such that, no matter how many nodes $n$ there are, we have that:

$$\mathbb{P}(|[\mathbf{y}_v]_i - \mathbb{E}[\mathbf{y}_v]_i| \geq t) = \mathbb{P}\left( \left| \sum_{u=1}^{n} ([\mathbf{w}_{u,v}]_i - \mathbb{E}[\mathbf{w}_{u,v}]_i) \right| \geq t \right) \leq 2 \exp\left( -\frac{ct^2}{K^2 n} \right)$$

Let's now compute $\mathbb{E}[\mathbf{y}_v]$, by first computing $\mathbb{E}[\mathbf{w}_{u,v}]$. When $u = v$ we have that:

$$\begin{aligned}
\mathbb{E}[\mathbf{w}_{v,v}] &= \mathbb{E}[(\boldsymbol{W}_s + \boldsymbol{W}_g)\mathbf{x}_v^{(0)}] \\
&= (\boldsymbol{W}_s + \boldsymbol{W}_g)\,\mathbb{E}[\mathbf{x}_v^{(0)}] \\
&= (\boldsymbol{W}_s + \boldsymbol{W}_g)\boldsymbol{\mu}
\end{aligned}$$

When $u \neq v$ we have, using the independence of $\mathbf{A}_{uv}$ and $\mathbf{x}_v$:

$$\mathbb{E}[\mathbf{w}_{u,v}] = \mathbb{E}([\mathbf{A}_{uv}\boldsymbol{W}_n + \boldsymbol{W}_g)\mathbf{x}_u^{(0)}]$$
$$= (\mathbb{E}[\mathbf{A}_{uv}]\boldsymbol{W}_n + \boldsymbol{W}_g)\,\mathbb{E}[\mathbf{x}_v^{(0)}]$$
$$= (r\boldsymbol{W}_n + \boldsymbol{W}_g)\boldsymbol{\mu}$$

Therefore (separating $\mathbf{w}_{v,v}$ from $\mathbf{w}_{u,v}$ for $u \neq v$):

$$\mathbb{E}[\mathbf{y}_v] = \sum_{u=1}^{n} \mathbb{E}[\mathbf{w}_{u,v}] + \boldsymbol{b} = (n-1)(r\boldsymbol{W}_n + \boldsymbol{W}_g)\boldsymbol{\mu} + (\boldsymbol{W}_s + \boldsymbol{W}_g)\boldsymbol{\mu} + \boldsymbol{b}$$

Since $\mathcal{M}$ is synchronously saturating for $\mathbb{G}(n,r)$ and $\mathbb{D}(d)$, we know that $[(r\boldsymbol{W}_n + \boldsymbol{W}_g)\boldsymbol{\mu}]_i \neq 0$. Assume without loss of generality that $[(r\boldsymbol{W}_n + \boldsymbol{W}_g)\boldsymbol{\mu}]_i > 0$. Then the expected value of $[\mathbf{y}_v]_i$ increases as $n$ tends to infinity; moreover we have a bound on how much $[\mathbf{y}_v]_i$ can vary around its expected value.

Recall that the non-linearity $\sigma$ is eventually constant in both directions. In particular, it is constant with value $\sigma_\infty$ above some $x_\infty$. When $\mathbb{E}[\mathbf{y}_v]_i > x_\infty$ the probability that $[\mathbf{y}_v]_i$ doesn't surpass this threshold is:

$$\mathbb{P}([\mathbf{y}_v]_i < x_\infty) \leq \mathbb{P}(|[\mathbf{y}_v]_i - \mathbb{E}[\mathbf{y}_v]_i| > |x_\infty - \mathbb{E}[\mathbf{y}_v]_i|)$$
$$\leq 2\exp\left(-\frac{c|x_\infty - \mathbb{E}[\mathbf{y}_v]_i|^2}{K^2 n}\right)$$

There is a constant $\rho$ such that $|x_\infty - \mathbb{E}[\mathbf{y}_v]_i| \geq \rho n$. Hence for sufficiently large $n$ (i.e. such that $\mathbb{E}[\mathbf{y}_v]_i > x_\infty$):

$$\mathbb{P}([\mathbf{y}_v]_i < x_\infty) \leq 2\exp\left(-\frac{c\rho^2 n^2}{K^2 n}\right) = 2\exp\left(-\frac{c\rho^2 n}{K^2}\right)$$

Since the activation $[\mathbf{x}_v]_i = \sigma([\mathbf{y}_v]_i)$, the probability that $[\mathbf{x}_v]_i$ takes value $\sigma_\infty$ is at least $1 - 2\exp\left(-c\rho^2 n/K^2\right)$. Now, for each node $v$ and each $i \in \{1, \ldots, d(1)\}$, the activation $[\mathbf{x}_v]_i$ is either $\sigma_\infty$ with high probability or $\sigma_{-\infty}$ with high probability. By taking a union bound, for sufficiently large $n$ the probability that every $[\mathbf{x}_v]_i$ takes its corresponding value is at least:

$$1 - 2nd(1)\exp\left(-\frac{c\rho^2 n}{K^2}\right)$$

This tends to 1 as $n$ tends to infinity. In other words, there is $\boldsymbol{z}_1 \in \{\sigma_{-\infty}, \sigma_\infty\}^{d(1)}$ such that $\mathbf{x}_v^{(1)} = \boldsymbol{z}_1$ for every $v$ asymptotically.

We now proceed to the second layer, and condition on the event that $\mathbf{x}_v^{(1)} = \boldsymbol{z}_1$ for every $v$. In this case, we have that the second layer preactivation for node $v$ is as follows.

$$\mathbf{y}_v^{(2)} = (\boldsymbol{W}_s^{(2)} + |\mathcal{N}(v)|\boldsymbol{W}_n^{(2)} + n\boldsymbol{W}_g^{(2)})\boldsymbol{z}_1 + \boldsymbol{b}^{(2)}$$

Since we're in the situation where every $\mathbf{x}_u^{(1)} = \boldsymbol{z}_1$, the degree $|\mathcal{N}(v)|$ is simply binomially distributed $\text{Bin}(n, r)$. The preactivation $\mathbf{y}_v^{(2)}$ then has expected value:

$$\mathbb{E}[\mathbf{y}_v^{(2)}] = n(r\boldsymbol{W}_n^{(2)} + \boldsymbol{W}_g^{(2)})\boldsymbol{z}_1 + \boldsymbol{W}_s^{(2)}\boldsymbol{z}_1 + \boldsymbol{b}^{(2)}$$

Fix $i \in \{1, \ldots, d(2)\}$. Since $\mathcal{M}$ is synchronously saturating for $\mathbb{G}(n,r)$ and $\mathbb{D}(d)$, we have that $[(r\boldsymbol{W}_n^{(2)} + \boldsymbol{W}_g^{(2)})\boldsymbol{z}_1]_i \neq 0$. Assume without loss of generality that $[(r\boldsymbol{W}_n^{(2)} + \boldsymbol{W}_g^{(2)})\boldsymbol{z}_1]_i > 0$. Then $\mathbb{E}[\mathbf{y}_v^{(2)}]_i$ tends to infinity as $n$ increases.

Furthermore, we can view $[n(r\boldsymbol{W}_n^{(2)} + \boldsymbol{W}_g^{(2)})\boldsymbol{z}_1]_i$ as the sum of $n$ Bernoulli random variables (which take values $[(\boldsymbol{W}_n^{(2)} + \boldsymbol{W}_g^{(2)})\boldsymbol{z}_1]_i$ and $[\boldsymbol{W}_g^{(2)}\boldsymbol{z}_1]_i$). Since by Lemma A.4 Bernoulli random variables are sub-Gaussian, as in the first-layer case we can apply Hoeffding's Inequality to bound the probability that $[\mathbf{y}_v^{(2)}]_i$ is less than $x_\infty$. We get that, for sufficiently large $n$, there is a constant $K$ such that this probability is bounded by $2\exp\left(-Kn\right)$.

Then, as before, we find $z_2 \in \{\sigma_{-\infty}, \sigma_\infty\}^{d(2)}$ such that, for sufficiently large $n$, every $\mathbf{x}_v^{(2)} = z_2$ with probability at least $1 - 2nd \exp(-Kn)$.

Finally, this argument is applied inductively through all layers. As the number of layers remains constant (since $\mathcal{M}$ is fixed), we find that the node embeddings throughout the model are asymptotically constant. $\qquad\square$

With the key lemma in place, we can now prove the main theorem.

*Proof of Theorem 4.10.* Applying Lemma 4.11 to the final layer, we find $z_T \in \{\sigma_{-\infty}, \sigma_\infty\}^{d(T)}$ such that every $\mathbf{x}_v^{(T)} = z_T$ with probability tending to 1. Since we use either average or component-wise maximum pooling, then means that the final graph-level representation is asymptotically constant, and thus the output of the classifier must be asymptotically constant. $\qquad\square$

# D   Proof of the uniform expressive power of SUMGNN⁺ with random features

We make use of a result due to Abboud et al. [1] which shows that SUMGNN⁺ models with random features can approximate any graph invariant on graphs with a fixed number of nodes.

**Definition D.1.** Let $f$ be a function on graphs, and let $\zeta$ be a random function on graphs. Take $\delta > 0$ and $N \in \mathbb{N}$. Then $\zeta$ *$\delta$-approximates $f$ up to $N$* if:

$$\forall n \leq N\colon \ \mathbb{P}(\zeta(G) = f(G) \mid |G| = n) \geq 1 - \delta$$

For completeness, we state the definition of the linearized sigmoid here.

**Definition D.2.** The *linearized* sigmoid is the function $\mathbb{R} \to \mathbb{R}$ defined as follows:

$$x \mapsto \begin{cases} -1 & \text{if } x \in (-\infty, -1), \\ x & \text{if } x \in [-1, 1), \\ 1 & \text{otherwise.} \end{cases}$$

**Theorem D.3.** *Let $\xi$ be any graph invariant. For every $N \in \mathbb{N}$ and $\delta > 0$ there is a SUMGNN⁺ with random features $\mathcal{M}$ which $\delta$-approximates $\xi$ up to $N$. Moreover, $\mathcal{M}$ uses the linearized sigmoid as the non-linearity and the distribution of the initial node embeddings consists of $d$ iid $U[0, 1]$ random variables.*

*Proof.* See [1, Theorem 1]. $\qquad\square$

With this result we can now prove the uniform expressivity result.

*Proof of Theorem 5.2.* First, $\xi$ satisfies a zero-one law for $\mathbb{G}(n, 1/2)$. Without loss of generality assume that $\xi$ is asymptotically 1. There is $N \in \mathbb{N}$ such that for every $n > N$ we have:

$$\mathbb{P}(\xi(G) = 1 \mid G \sim \mathbb{G}(n, 1/2)) \geq 1 - \delta$$

Note that this $N$ depends on both $\xi$ and $\delta$.

Second, by Theorem D.3 there is a SUMGNN⁺ with random features $\mathcal{M}$ which $\delta$-approximates $\xi$ up to $N$. Moreover, $\mathcal{M}'$ uses the linearized sigmoid as the non-linearity and the distribution of the initial node embeddings consists of $d$ iid $U[0, 1]$ random variables.

Using the global readout and the linearized sigmoid, we can condition the model behavior on the number of nodes. We give a rough description of the model as follows. Define a SUMGNN⁺ with random features $\mathcal{M}$ by extending $\mathcal{M}'$ as follows.

- Increase the number of layers to at least three.

- Increase each embedding dimension by 1. For convenience call this the 0th component of each embedding.

- Use the bias term in the first layer to ensure that the 0th component of the activation $\mathbf{x}_v^{(1)}$ for each node $v$ is 1.

- Use the global readout to threshold the number of nodes on $N$. The 0th row of the matrix $\boldsymbol{W}_g^{(2)}$ should have a 2 in the 0th position and 0's elsewhere. The 0th component of the bias vector $\boldsymbol{b}^{(2)}$ should be $2N - 1$. This ensures that the 0th component of every activation $\mathbf{x}_v^{(2)}$ is 1 if $n > N$ and $-1$ otherwise.

- Propagate this value through the 0th component of each layer embedding.

- In the final layer, use this value to decide whether to output what $\mathcal{M}'$ would output, or simply to output 1.

For any $n \leq N$ the model $\mathcal{M}$ behaves like $\mathcal{M}'$. Therefore:
$$\mathbb{P}(\xi(G) = \mathcal{M}(G) \mid |G| = n) \geq 1 - \delta$$
On the other hand, for $n > N$ the model $\mathcal{M}$ simply outputs 1 and so:
$$\mathbb{P}(\xi(G) = \mathcal{M}(G) \mid |G| = n) = \mathbb{P}(\xi(G) = 1 \mid |G| = n) \geq 1 - \delta$$
Thus $\mathcal{M}$ uniformly $\delta$-approximates $\xi$. $\qquad\square$

# E    Further Experiments

In this section, we focus on GCNs and provide further experiments regarding our results. In particular, we pose the following questions:

1. Our theoretical results entail a zero-one law for a large class of distributions: do we empirically observe a zero-one law when node features are instead drawn from a normal distribution (Appendix E.1)?

2. Our theoretical results state a zero-one law for a large class of non-linearities: do we empirically observe a zero-one law when considering other common non-linearities (Appendix E.2)?

3. Does a zero-one law also manifest itself empirically for GAT models (Appendix E.3)?

4. Do we empirically observe a zero-one law if we were to consider sparse Erdős-Rényi graphs (Appendix E.4)?

5. Is there empirical evidence for our results to apply to other random graph models, such as the Barabási-Albert model (Appendix E.5)?

## E.1    Experiments with initial node features drawn from a normal distribution

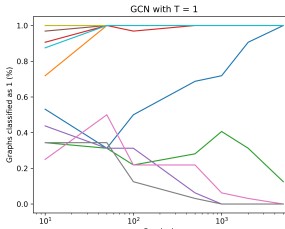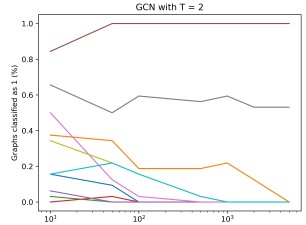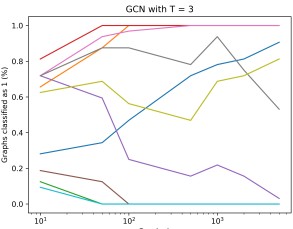

Figure 2: Normally distributed random node features with GCN models. Each plot shows the proportion of graphs of certain size which are classified as 1 by a set of ten GCN models. Each curve (color-coded) shows the behavior of a model, as we draw increasingly larger graphs. The phenomenon is observed for 1-layer models (left column), 2-layer models (mid column), and 3-layer models (last column). We draw the initial features randomly from a normal distribution with mean 0.5 and standard deviation 1.

Here we consider using a normal distribution to draw our initial node features. Note that normal distributions are sub-Gaussian, and hence our theoretical findings (Theorem 4.6) confer a zero-one law in this case. Figure 2 demonstrates the results for GCN models. We observe the expected asymptotic behavior in most cases, however in the two and three layer cases a few models have not converged by the end of the experiment.

## E.2 Experiments with other non-linearities

In this subsection we test the effect of using different non-linearities in the layers of our GCN models. Theorem 4.6 applies in all of these cases, so we do expect to see a zero-one law. Figures 3 to 5 present the results for ReLU, tanh and sigmoid, respectively. We see the expected behavior in all cases. Note however that, in contrast with other non-linearities, when we use sigmoid we observe that the rate of convergence actually increases as the number of layers increases. This suggests a complex relationship between the rate of convergence, the non-linearity and the number of layers.

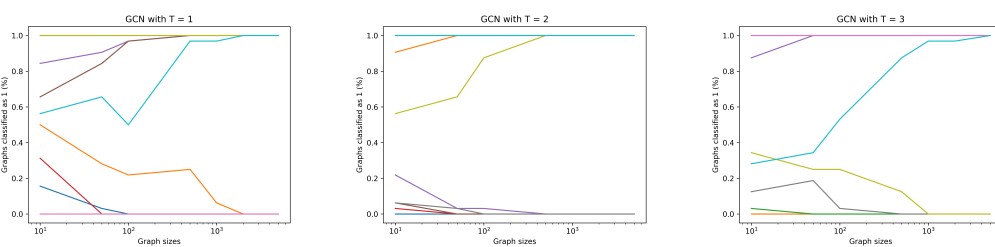

Figure 3: GCN models with ReLU non-linearity. Each plot shows the proportion of graphs of certain size which are classified as 1 by a set of ten GCN models. Each curve (color-coded) shows the behavior of a model, as we draw increasingly larger graphs. The phenomenon is observed for 1-layer models (left column), 2-layer models (mid column), and 3-layer models (last column). This time we choose the ReLU activation function for the GNN layers. Apart from this, the setup is the same as in the main body of the paper.

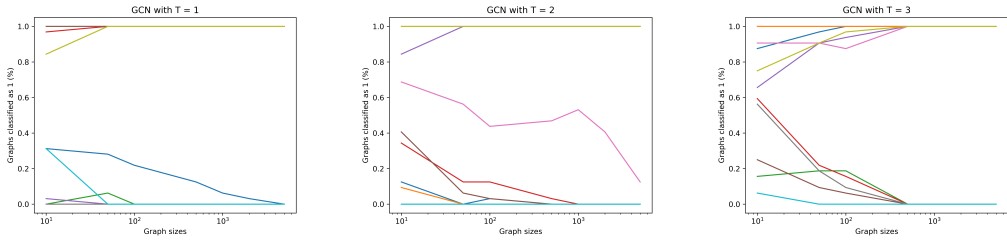

Figure 4: GCN models with tanh non-linearity. Each plot shows the proportion of graphs of certain size which are classified as 1 by a set of ten GCN models. Each curve (color-coded) shows the behavior of a model, as we draw increasingly larger graphs. The phenomenon is observed for 1-layer models (left column), 2-layer models (mid column), and 3-layer models (last column). We use tanh as an activation function for the GNN layers, and keep everything else the same.

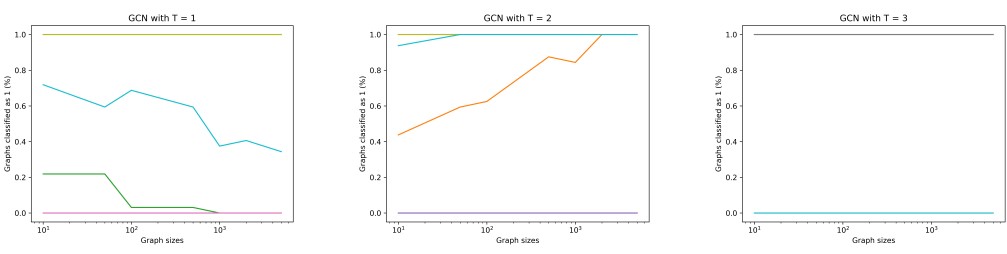

Figure 5: GCN models with sigmoid non-linearity. Each plot shows the proportion of graphs of certain size which are classified as 1 by a set of ten GCN models. Each curve (color-coded) shows the behavior of a model, as we draw increasingly larger graphs. The phenomenon is observed for 1-layer models (left column), 2-layer models (mid column), and 3-layer models (last column). We use the sigmoid activation function for the GNN layers, and keep everything else the same.

### E.3 Experiments with GAT

Here we investigate the asymptotic behaviour of a GNN architecture not considered in the main body: the Graph Attention Network [37]. Cast as an MPNN, this architecture uses an attention mechanism as the aggregate function $\phi$ in the message passing step. The techniques used in this paper to establish a zero-one law for other GNN architectures do not easily extend to GAT. However, our experiments demonstrate a very quick convergence to $0$ or $1$.

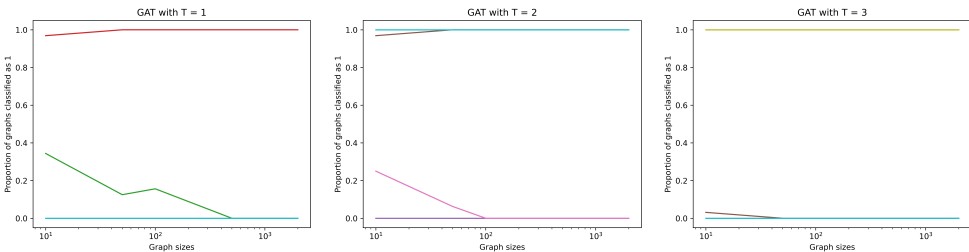

Figure 6: Ten GAT models with number of layers 1, 2 and 3 are run on graphs of increasing size, with the proportion of nodes classified as 1 recorded. We observe a convergence to zero-one law very quickly.

### E.4 Experiments on sparse Erdős-Rényi graphs

In these experiments, we consider GCN models on a variation of the Erdős-Rényi distribution in which the edge probability $r$ is allowed to vary as a function of $n$. Specifically, we set $r = \log(n)/n$, which yields sparser graphs than in the standard distribution. Our experiments provide evidence for a zero-one law also in this case (Figure 7).

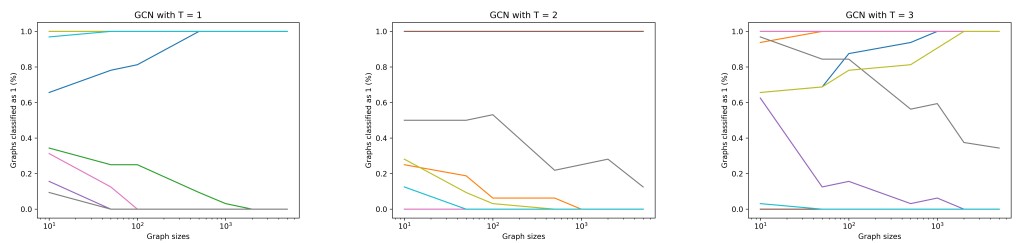

Figure 7: Sparse Erdős-Rényi graphs with GCN models. Each plot shows the proportion of graphs of certain size which are classified as 1 by a set of ten GCN models. Each curve (color-coded) shows the behavior of a model, as we draw increasingly larger graphs. The phenomenon is observed for 1-layer models (left column), 2-layer models (mid column), and 3-layer models (last column). We let the probability $r$ of an edge appearing be $\frac{\log(n)}{n}$. All the other parameters are the same as in the experiments of the main body of the paper.

### E.5 Experiments on the Barabási-Albert random graph model

In this subsection, we consider another alternative graph distribution: the Barabási-Albert model [3]. This model aims to better capture the degree distributions commonly found in real-world networks. We can again observe a zero-one law for GCN models under this distribution (Figure 8).

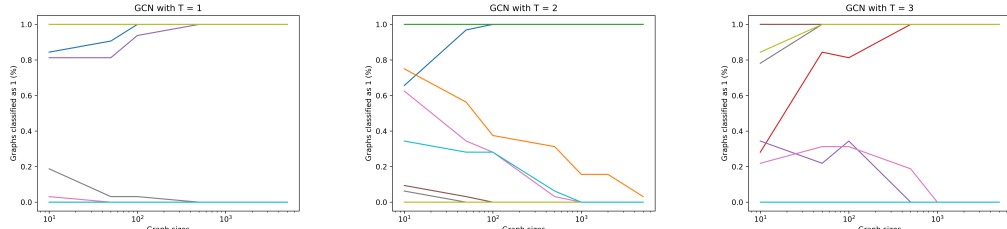

Figure 8: Barabási-Albert graphs with GCN models. Each plot shows the proportion of graphs of certain size which are classified as 1 by a set of ten GCN models. Each curve (color-coded) shows the behavior of a model, as we draw increasingly larger graphs. The phenomenon is observed for 1-layer models (left column), 2-layer models (mid column), and 3-layer models (last column). We generate the graphs using the Barabási-Albert model; apart from this the setup is the main as in the experiments in the main body of the paper.

