# OpenReview forum: "Zero-One Laws of Graph Neural Networks"
_NeurIPS.cc/2023/Conference — NeurIPS 2023 poster_

### Official Review · Reviewer_P48L · 2023-07-04

**Soundness:** 4 excellent
**Presentation:** 3 good
**Contribution:** 3 good
**Rating:** 6
**Confidence:** 3

**Summary:**

The authors analyze standard GNN binary classifiers on Erdos-Renyi graphs, and conclude that the output satisfies a zero-one law. To be more precise, this means that an Erdos-Renyi graph with the number of nodes tending to infinity and random node features will be classified as 1 with probability tending towards either 0 or 1. Of course, this comes with some assumptions, which the authors claim are mild and reasonable to assume. The authors also provide empirical data that suggests n does not need to be incredibly large (on the order of 1,000 or 10,000) for this effect to take hold.

**Strengths:**

The strengths of this paper lie in the novelty of the authors' result. The result is interesting and significant to the field of investigating the theoretical upper and lower bound of expressivity in GNNs, which will inspire other readers to expand upon the investigation, particularly in zero-one laws. Experiments done are also convincing in supporting the practicality of their argument and to give readers a visual indication of the authors' result.

**Weaknesses:**

The paper's main weakness lies in their dismissal of assumptions. Although the result is interesting, its overall current practicality may be somewhat overstated. For example, in line 173 authors claim sub-Gaussian random vectors "encompass all practical setups." This may be quite a bold statement, as many real-world examples are known to follow common distributions such as the exponential or Pareto distributions, and are fitting for further investigation. In addition, the same can be said about the ER random graph model; it is useful as an analytical tool, but fits to real graph topologies rather poorly. Perhaps more theoretical work or empirical investigation can be done on both fronts. (I acknowledge that there are empirical results on the Barabási-Albert model, which is helpful.)

If treated as a purely theoretical result/paper though, these addressed weaknesses may be taken lightly.

Also, would it be possible to add confidence bounds for the graphs in this case?

**Questions:**

1. Do the proofs extend easily to different random graph structures and/or wider families of distributions?
2. Why was it not possible to run the Figure 1 plot for larger graphs? SUMGNN T=3 seems like it may converge, but it would be nice to see it visually.


**Limitations:**

Limitations address key points, although could be elaborated upon more. The current version may be toning down the limitations as addressed above in the weakness section.

No potential negative societal impact.

---

> ### Author Rebuttal · Authors · 2023-08-09
>
> Thank you for your comments on our paper, which we address below.
>
> > Do the proofs extend easily to different random graph structures and/or wider families of distributions?
>
> We haven’t found an easy extension to different distributions of graphs with node features. Once edges and features become dependent, it becomes harder to apply classical concentration inequalities, and so new techniques would be needed in this case. We note however that in the classical literature on first-order logic, to obey a zero-one law means to do so with respect to ER graphs.
>
> > Why was it not possible to run the Figure 1 plot for larger graphs? SUMGNN T=3 seems like it may converge, but it would be nice to see it visually.
>
> The principal reason for not continuing the graph was a limitation of our infrastructure setup. We have now rerun the experiment up to larger graphs, which indeed show convergence around $10^5$ nodes. We include the figure in our global response PDF and will replace the corresponding figure in the paper.
>
> > Also, would it be possible to add confidence bounds for the graphs in this case?
>
> Thank you for your suggestion. We would like to incorporate it into our paper, but we noticed it may hinder the clarity of the presentation, because of the following reason. Because the asymptotic behaviour goes one of two ways, to add confidence bounds we need need to either separate the data into ‘asymptotically 0’ and ‘asymptotically 1’ by inspection, or else perform some transformation on the data (such as $x \mapsto 2 |x - 0.5|$), which would reduce the demonstrative force of the figures. If you think the confidence bounds could be nonetheless helpful, we will do our best to integrate these into our presentation.
>
> > Although the result is interesting, its overall current practicality may be somewhat overstated. For example, in line 173 authors claim sub-Gaussian random vectors "encompass all practical setups." This may be quite a bold statement, as many real-world examples are known to follow common distributions such as the exponential or Pareto distributions, and are fitting for further investigation.
>
> We wish to note that, while real world examples may have distributions which are not sub-Gaussian, as you mention, in practice all node features will have bounded values (determined by the bit length of the storage medium), and are thus sub-Gaussian. We’ll add a sentence to explain this.

---

> > ### Comment · Reviewer_P48L · 2023-08-14
> >
> > I thank the authors for their detailed answers and addressing my misunderstanding. I will retain my score for the time being.

---

> > > ### Author Response · Authors · 2023-08-14
> > > **Reply**
> > >
> > > Thank you for your comment! Please let us know if any further improvements or clarifications are needed. We wish to do our best to clearly convey the message of the paper.

---

### Official Review · Reviewer_Mx1j · 2023-07-06

**Soundness:** 3 good
**Presentation:** 3 good
**Contribution:** 3 good
**Rating:** 7
**Confidence:** 4

**Summary:**

This paper is primarily a theory paper - proposing a proof that, as the number of nodes in a graph drawn from the ER graph distribution increases, a class of GNNs binary classifiers converges to either 1 or 0.

This observation is backed up with a small set of empirical experiments of GCNs and variants, but no additional experiments on more advanced GNNs.



**Strengths:**

I thought the paper was well written, and the mathematics was clearly presented.

The observations in the text were illuminating and intution was added at appropriate points. I felt that the exposition was not overly confusing, as can be the case with a number of theory papers, and the results put in context. Overall, I really enjoyed reading this, which is a rarity for theory heavy papers.

**Weaknesses:**

I don't think it can be argued that it is a weakness that the paper's experimental results are for a class of models that isn't very commonly used any more - GCNs as they are presented in this work have been superseded by any number of architectural innovations. There are not, for example, experiments or proofs on the impact of residual layers or skip connections, or GNNs with edge MLPs. As such, I think while it's a very nice result, the impact of the paper is likely to be low.

It wasn't clear to me - and I think this is because the math would be impossible to do for every architecture - how this result depends upon the scale seen in a number of graph neural networks, or how it changes if you add an attention mechanism. It could be that, in practice, this effect is only relevant if you have a very underpowered network, or you have a very very large graph. Both of which are not hugely practical examples.

**Questions:**

How would an attention aggregation change the analysis? For me, that would hugely improve the impact of the paper because you're heading towards transformers at that point and so a far wider audience.

**Limitations:**

Limitations have been expressed above.

---

> ### Author Rebuttal · Authors · 2023-08-09
>
> Thank you very much for your review, and your comments on the clarity of the exposition.
>
> > How would an attention aggregation change the analysis? For me, that would hugely improve the impact of the paper because you're heading towards transformers at that point and so a far wider audience.
>
> The study of attention in its full generality is very interesting, particularly in connection to transformers, but also very challenging, because it can capture wildly different behaviours. On the one hand, we have found that if we restrict the attention to follow a distribution which closely follows a Dirichlet distribution (or, deviates from mean-aggregation with some bounded epsilon), then our techniques for MeanGNNs will apply with minor modifications. On the other hand, in practice attention is conditioned on the feature of the source node and so it can act non-uniformly depending on the source node, effectively emulating different aggregations for different nodes.
>
> Under the light of these considerations, we view a fine-grained analysis of attention models under different assumptions as a challenging future work. We have, however, conducted experimental analysis on GATs, which do appear to display a zero-one law quite strongly; we conjecture this is due to the static attention of GATs, which is shown to be a restricted form of attention (Brody et al, 2022). We include this experimental result in our 'global' response PDF.

---

> > ### Comment · Reviewer_Mx1j · 2023-08-17
> >
> > Thanks for your rebuttal! Your comments make sense, transformers are weird beasts.
> >
> > Still think this is a good paper, will keep my score the same.

---

> > > ### Author Response · Authors · 2023-08-17
> > >
> > > Thank you for going through the rebuttal, and for your continued positive evaluation of our paper.

---

### Official Review · Reviewer_ye6a · 2023-07-07

**Soundness:** 4 excellent
**Presentation:** 3 good
**Contribution:** 4 excellent
**Rating:** 6
**Confidence:** 4

**Summary:**

This paper introduces the zero-one laws pertaining to graph neural networks. More specifically, the authors utilize concentration analysis on Erdos-Renyi graphs to demonstrate that the likelihood of GNN classifiers producing either zero or one as output approaches 1. These findings are further expanded to encompass three different methods of aggregation and normalization in GNNs. Additionally, the authors present uniform approximation results for graph invariants that adhere to the zero-one law.

**Strengths:**

+ The primary assertion is intriguing and offers a fresh perspective for evaluating the expressive capabilities of graphs.

+ This paper presents valid theoretical contributions, establishing the first asymptotic analysis of GNN behavior as the number of nodes approaches infinity. This analysis yields negative outcomes regarding GNN expressiveness, which may prompt reconsideration of new GNN architecture designs.

+ Despite its theoretical soundness, the paper also conducts well-designed experiments to substantiate its claims.


**Weaknesses:**

- Certain theoretical definitions and arguments lack intuitive explanations, making it challenging to justify the assumptions. (See Questions)

- To the best of my knowledge, global readout typically occurs only in the last layer. However, it appears that both MeanGNN and SumGNN incorporate it in each layer. Although this may not necessarily complicate the analysis, it may deviate from practical usage.

- Are the arguments limited to ER graphs and sub-gaussian features? Does the concentration primarily stem from the assumptions regarding ER graphs and sub-gaussian features, rather than establishing a unique zero-one law specific to GNNs? (See Questions)


**Questions:**

1. I still lack access to the specific paper mentioned, so I can offer only general suggestions for rephrasing the paragraph based on the information provided:
I find it challenging to fully comprehend the correct interpretation of Definition 4.9, "synchronously saturating." In addition to presenting reasonable concentration results in the proof, it would be beneficial if the authors could provide a more straightforward explanation. Furthermore, I would appreciate further justification for the claim made: (Ln 242-243) In fact, assuming that µ, σ−∞ and σ∞ are non-zero, the the set of weights which yield a non-synchronously-saturating model has measure zero?

2. If I understand correctly, Section 5 primarily argues that the zero-one law is not always disadvantageous, as it enables uniform approximation. However, I'm curious why approximating a function with a fixed output (either 0 or 1) is considered desirable?

3. The paper uncovers an intriguing zero-one law emerging in GNNs. However, I would like to inquire whether the ER graph and sub-gaussian feature assumptions are the primary factors contributing to this phenomenon. It seems that the concentration is not a result of any specific components within GNNs, but rather an outcome of the assumed input.

4. Could the authors shed light on how the asymptotics correspond to the number of layers? Do additional layers lead to faster or slower concentration?


**Limitations:**

The paper addresses certain limitations, and in addition to those already discussed, I recommend that the authors include a discussion on the strict assumptions made. Examining the impact of removing either the ER graph assumption or the sub-gaussian condition could potentially yield more substantial results.

---

> ### Author Rebuttal · Authors · 2023-08-09
>
> Thank you for your review and for your comments. We address your concerns below.
>
> > I still lack access to the specific paper mentioned, so I can offer only general suggestions for rephrasing the paragraph based on the information provided: I find it challenging to fully comprehend the correct interpretation of Definition 4.9, "synchronously saturating." In addition to presenting reasonable concentration results in the proof, it would be beneficial if the authors could provide a more straightforward explanation. Furthermore, I would appreciate further justification for the claim made: (Ln 242-243) In fact, assuming that µ, σ−∞ and σ∞ are non-zero, the the set of weights which yield a non-synchronously-saturating model has measure zero?
>
> Analysis of the proof of the main result reveals that the asymptotic behaviour is determined by the matrices $Q_t = rW_n^{(t)} + W_g^{(t)}$, where asymptotic final layer embeddings are $\sigma(Q_T(\sigma(Q_{T-1} \cdots \sigma(Q_0 \mu) \cdots ))))$. To be synchronously saturating is to avoid the boundary case where one of the intermediate steps in the asymptotic final layer embedding computation has a zero component.
>
> This is expressed by finitely many linear non-equalities. The space of models which is not synchronously saturating is the union of solution space of each equality, each of which space has dimension lower than the whole space. The space of such models is thus of lower dimension, and therefore of zero measure.
>
> We will add this explanation to the paper.
>
> > If I understand correctly, Section 5 primarily argues that the zero-one law is not always disadvantageous, as it enables uniform approximation. However, I'm curious why approximating a function with a fixed output (either 0 or 1) is considered desirable?
>
> In Section 5 we aim for a more complete characterisation of the expressive power of our GNNs, by establishing a partial matching lower bound for the uniform expressivity upper bound established in the previous section. Informally, this corresponds to answering the following complementary question: knowing that we can ‘at best’ capture properties which follow a zero-one law, is it possible to capture all such properties with GNNs? For this theoretical result, given any property which satisfies a certain zero-one law we need to find some GNN model which uniformly approximates it.
>
> > The paper uncovers an intriguing zero-one law emerging in GNNs. However, I would like to inquire whether the ER graph and sub-gaussian feature assumptions are the primary factors contributing to this phenomenon. It seems that the concentration is not a result of any specific components within GNNs, but rather an outcome of the assumed input.
>
> Fundamentally, zero-one laws are properties of languages (or classes of models) not the respective inputs. Indeed, the fact that the sum-aggregation and mean-aggregation models require different techniques to establish a zero-one law suggests the choice of aggregation component is an important factor contributing to the law. We would like to draw a comparison with analogous zero-one laws established for first-order formulas by Glebskii et al., (1969) and Fagin (1976). These classical findings are generally regarded as establishing a surprising property of first-order logic, rather than of ER graphs. Our work is in exactly the same spirit: we use ER graphs as a distribution to examine the class of properties expressible by GNN models, just as they have been used to investigate the class of first-order properties.
>
> We note also that in practice all node features will have bounded values (determined by the bit length of the storage medium), and are thus sub-Gaussian. Therefore this assumption is a very mild one.
>
> > Could the authors shed light on how the asymptotics correspond to the number of layers? Do additional layers lead to faster or slower concentration?
>
> A formal analysis of the rate of convergence would be quite non-trivial. The natural way to approach this would be to keep track of the variance of the node representations throughout the network. However, there are a number of dependencies between these, which would necessitate computing the covariance matrices, which complicates the analysis.
>
> In the appendix we include experiments investigating the effect of using different nonlinearities. We have found that while for some choices of nonlinearity convergence is slower for a greater number of layers, for other choices the opposite phenomenon occurs (see Appendix E.2). We therefore expect a complex picture, in which the rate of convergence depends on the choice of nonlinearity, the number of layers and (likely) the width of those layers.
>
> > To the best of my knowledge, global readout typically occurs only in the last layer. However, it appears that both MeanGNN and SumGNN incorporate it in each layer. Although this may not necessarily complicate the analysis, it may deviate from practical usage.
>
> We note that our results apply equally to the case where the global readout occurs in the last layer, or indeed not at all, by setting the global readout weights to 0. We allow a global readout in all layers to make our results more general.

---

> > ### Comment · Reviewer_ye6a · 2023-08-21
> > **Thanks for rebuttal!**
> >
> > I appreciate authors' reply which addresses all my concerns. I value the topics and conclusions drawn in this paper and agree with authors that ER graphs are good examples to investigate, aligned with classic results. In the meanwhile, the discussion on this assumption has already been clarified in the limitation section (Sec. 7), which I do not think can be used to overturn this work itself. In summary, I'm lean toward acceptance.

---

> > > ### Author Response · Authors · 2023-08-21
> > >
> > > Thank you for going through the rebuttal, and for your positive evaluation of our paper. We are happy that our rebuttal addressed all your concerns and that you value the topics and conclusions drawn in our paper. Thanks also for raising the point regarding the use of ER graphs and its alignment with the classical literature, very much appreciated.

---

### Official Review · Reviewer_BoTS · 2023-07-12

**Soundness:** 3 good
**Presentation:** 3 good
**Contribution:** 2 fair
**Rating:** 3
**Confidence:** 4

**Summary:**


This paper proves that for Erdos-Renyi random graphs, the hidden representations of nodes for GNNs, as well as the graph-level representation, will converge to a constant. In Theorem 4.6, which is the main result for GCN and binary graph classification, they prove that the binary output is always zero or one (the zero-one law). This means that the classifier will eventually become trivial. Its proof is based on Lemma 4.7, which shows the concentration of hidden representations on nodes. They then extend their theory in Theorem 4.10 to SumGNNs and show (in Theorem 5.2) that indeed zero-one laws imply uniform approximation by SumGNNs, which can be thought of as a converse to their zero-one law results. The paper is concluded with supporting experiments.



**Strengths:**

- nice theoretical results
- citing previous works


**Weaknesses:**

- the results are restricted to ER graphs, which are not what we have in practice. Also, their model is restricted to dense graphs.
- the paper is generally well-written, but some parts are long and could be more concise (like the meaning of the zero-one law and its motivation); they could've used the space better




**Questions:**



I believe this is an interesting result. But unfortunately, it is far from practical models for GNNs. Here I list my comments:




Major comments:



(0) One major problem regarding the writing of the paper is that many paragraphs try to deliver the simple concept of the zero-one law for GNNs and its importance/motivation, which was too long for me. The paper is not concise.

(1) In the paper, it is mentioned that this is a well-known result: "any graph invariant which can be expressed by a first-order formula satisfies a zero-one law" How does this relate to your result? Are you claiming that non-first-order formulas are satisfying the zero-one law? Otherwise, it falls in that category and it's not a new result. If not, an extensive explanation is required for how your proof is different and how those results are connected in each other.

(2) line 209: "we expect.." This is indeed an important problem that is left unanswered. The rate of convergence shows when GNNs are not useful, so it may be exponentially slow in the number of nodes.

(3) Many results in the paper are conditional on excluding an unknown zero-measure set of parameters. However, optimization algorithms (like SGD) trajectory is also a set of measure zero. The local minima of the objective may fall in those sets.

 (4) In Lemma 4.11, it is proved that the hidden representations of nodes, for any iteration, will converge to a fixed value. This is highly restricted by the assumption that the non-linearity saturates. As a result, it will not directly apply to ReLU networks.




(5) The results are highly restricted. Random initial features are drawn based on the same distribution across different nodes. They cannot depend on the graph. The ER model is itself not realistic because we hope GNNs can extract hidden features from the graph data, while random ER graphs don't have any meaningful "feature." ER graphs are dense, while in practice, sparse graphs (like grids) also appear. Besides that, the paper proofs apply after using the concentration inequalities for a large number of nodes. It thus looks reasonable to expect those zero-one laws to happen.



Minor comments:


 (0) line 67 - are the initial features and the random graph independently drawn? Or they can depend on each other?  (it is later mentioned in Definition 4.2) Also, it is unclear whether the random initial features of different nodes are independent.

 (1) line 73 - can the functions $\psi$ and $\phi$ depend on the iteration index $t$? This is not clearly mentioned there.

(2) line 263: typo: "every node $u$" -> "every node $v$"

---

> ### Author Rebuttal · Authors · 2023-08-09
>
> Thank you for your review and your comments on our paper. We address each below.
>
> > (0) One major problem regarding the writing of the paper is that many paragraphs try to deliver the simple concept of the zero-one law for GNNs and its importance/motivation, which was too long for me.
>
> To the best of our knowledge, using a zero-one law to give an upper bound on uniform expressivity is a new technique in (graph) ML, though well studied in mathematical logic. While the concept of a zero-one law is not too complex, the perspective of using it to characterise the class of properties which GNNs can capture is new, and thus requires more careful explanation. This motivation is also essential to understand why the choice of ER model is a reasonable one, because it establishes a result on uniform expressivity in line with similar results for first-order logic. We are happy to make all these explanations more concise in line with your request. Could you please let us know which parts/sections could be improved in this respect?
>
> > (1) In the paper, it is mentioned that this is a well-known result: "any graph invariant which can be expressed by a first-order formula satisfies a zero-one law [...]
>
> GNNs can express logical formulas in certain two-variable fragments of first-order logic (Barcelo et al 2020), but the upper bound remains open: there is *no* result which states that every GNN expresses a first-order formula; conversely, it is known that GNNs cannot express every first-order formula. Hence, even though related, there is no implication between Fagin’s result and ours. Our result further helps with the quest of identifying the precise logical fragment the studied GNNs can capture: the logic that can be captured by these GNNs must follow a zero-one law. We note that Fagin’s proof for first-order formulas works in a completely different way, relying on logical techniques, whereas our techniques are more probabilistic.
>
> > (2) line 209: "we expect.." This is indeed an important problem that is left unanswered. [...]
>
> We agree that the study on the rate of convergence is important, but given its complexity, it deserves a separate study. The natural way to approach this would be to keep track of the variance of the node representations throughout the network. However, there are a number of dependencies between these, which would necessitate computing the covariance matrices, which complicates the analysis.
>
> That being said, we do not think this presents a weakness for our main question: what class of properties can GNNs (uniformly) express? What we have shown is that if a GNN expresses a property $P$, then $P$ must satisfy a zero-one law on ER graphs. This limits the expressivity of these models to this class of properties. For example, as far as we are aware this is the *first result* showing that GNNs cannot express node-number parity. Please note that, for the expressiveness study, we don’t need any bounds on convergence, as it holds regardless. We nonetheless conjecture a fast convergence, as indicated in the experimental results.
>
> > (3) Many results in the paper are conditional on excluding an unknown zero-measure set of parameters. [...]
>
> Could you explain why the fact that the trajectory is of zero measure makes it plausible that a local minimum falls in any other particular set of zero-measure? The excluded set is a fixed zero-measure (in fact lower dimensional) subset of the parameter space. The optimization trajectory is one-dimensional, so is unlikely to even intersect any set of zero-measure, let alone finish there.
>
> > (4) In Lemma 4.11, it is proved that the hidden representations of nodes, for any iteration, will converge to a fixed value. This is highly restricted by the assumption that the non-linearity saturates.
>
> We acknowledge this as a limitation for our result on SumGNN models, but note that the result still applies to some common non-linearities, such as the sigmoid function (under fixed precision), and that this limitation does not apply to the GCN and MeanGNN results.
>
> We include in our global response PDF an experiment on the asymptotic behaviour of ReLU SumGNN models. This suggests that a zero-one law holds in practice. However, our techniques do not straightforwardly extend to the ReLU case, and this would require further study.
>
> > (5) The results are highly restricted. Random initial features are drawn based on the same distribution across different nodes. [...]
>
> In most theoretical studies we are aware of, it is standard to assume node features from the same distribution, and we align with these studies. It is true that graph topology and node features can be dependent, but the degree of this dependence is hard to determine and could be very different depending on the domain, and on other factors, e.g., whether graphs are homophilic or heterophilic. Hence, we think assuming independence is a reasonable choice for theoretical investigations.
>
> But let us emphasise that the realisticness of ER graphs has no bearing on the validity of the result establishing a uniform bound on expressivity. In fact an analogous result with respect to *any* graph distribution would provide a new bound (i.e. that properties captured by GNNs must satisfy a zero-one law with respect to that distribution). The choice of ER graphs is common in similar studies on zero-one laws and also in related works in the graph ML community; see, e.g., Oono et al (2020) which analyses the over-smoothing phenomenon of GNNs with respect to ER graphs. We therefore consider it an appropriate choice for our analysis.
>
> > (0) line 67 - are the initial features and the random graph independently drawn? [...]
>
> We will add a sentence clarifying this part: random initial features are independent both from each other and the graph adjacency matrix.
>
> > (1) line 73 - can the functions ψ and φ depend on the iteration index ?
>
> The functions can indeed vary between iterations. We will add an index to make this explicit.

---

> > ### Comment · Reviewer_BoTS · 2023-08-14
> > **Response**
> >
> > I acknowledge the response provided by the authors. However, I still disagree with the authors on the use of ER model (and other assumptions) for the theoretical analysis of GNNs. Unfortunately, I believe that the model, while being nice for theoretical analysis, is not appropriate for the study of GNNs.
> >
> > I decided to keep my score unchanged.

---

> > > ### Author Response · Authors · 2023-08-14
> > >
> > > We strongly disagree with the reviewer and have difficulty in understanding their justification for the provided score. We consider it inadequate to dismiss the work based on the use of ER graphs: in the classical literature on first-order logic, to obey a zero-one law means to do so with respect to ER graphs and our study aligns with this. We provide the first upper bound on the uniform expressiveness of GNNs, which paves the way for the use of zero-one laws in expressiveness studies of GNNs. We kindly ask the reviewer to revisit their evaluation based on this.

---

### Author Rebuttal · Authors · 2023-08-09

We wish to thank the reviewers for their comments. We have responded to each concern in detail in our individual responses. In addition, we include a PDF containing the results of additional experiments as suggested by reviewer comments.

Here is a summary of the changes to be made to the paper in light of the reviews received.
- We will include the extended SumGNN experiment (Figure 1 in the PDF) in place of the original. We will also include the additional experiments in the appendix.
- We will add a number of clarifying sentences, as suggested by reviewer comments.
- A few typos and minor errors will be corrected.

We look forward to continuing the discussion with the reviewers in the coming days.

---

### Decision · Program_Chairs · 2023-09-21

**Decision:**

Accept (poster)

**Comment:**

This paper analyzes standard GNN binary classifiers on Erdos-Renyi graphs, and identifies that the output satisfies a zero-one law: an Erdos-Renyi graph with the number of nodes tending to infinity and random node features will be classified as 1 with probability tending towards either 0 or 1.

Although reviewers have concerns about the assumptions, the results presented in this paper is interesting and significant to the community investigating the theoretical upper and lower bound of expressivity in GNNs, which may prompt new GNN architecture designs. Experiments presented in this paper are convincing as well.